

# Development and assessment of uni- and multi-variable flood loss models for Emilia-Romagna (Italy)

Francesca Carisi[1], Kai Schröter[2], Alessio Domeneghetti[1], Heidi Kreibich[2], and Attilio Castellarin[1]

[1]University of Bologna, DICAM, Water Resources, Bologna, Italy
[2]Hydrology Section, German Research Centre for Geosciences, GFZ, Potsdam, Germany

*Correspondence to:* Francesca Carisi (francesca.carisi@unibo.it)

**Abstract.**

Simplified flood loss models are one important source of uncertainty in flood risk assessments. Many countries experience sparseness or absence of comprehensive high-quality flood loss data sets which is often rooted in a lack of protocols and reference procedures for compiling loss data sets after flood events. Such data are an important reference for developing and validating flood loss models. We consider the Secchia river flood event of January 2014, when a sudden levee-breach caused the inundation of nearly 52 km$^2$ in Northern Italy. For this event we compiled a comprehensive flood loss data set of affected private households including buildings footprint, economic value, damages to contents, etc. based on information collected by local authorities after the event. By analysing this data set we tackle the problem of flood damage estimation in Emilia-Romagna (Italy) by identifying empirical uni- and multi-variable loss models for residential buildings and contents. The accuracy of the proposed models is compared with those of several flood-damage models reported in the literature, providing additional insights on the transferability of the models between different contexts. Our results show that (1) even simple uni-variable damage models based on local data are significantly more accurate than literature models derived for different contexts; (2) multi-variable models that consider several explanatory variables outperform uni-variable models which use only water depth. However, multi-variable models can only be effectively developed and applied if sufficient and detailed information is available.

## 1 Introduction

According to analyses of the Centre for Research on the Epidemiology of Disasters - CRED, hydrological disasters (i.e., natural disasters caused by river and coastal floods, flash-floods, rainstorms, etc.) are the most frequently recorded natural calamities occurring worldwide in the last two decades (see e.g. Guha-Sapir and CRED, 2015). Also, the number of disasters caused by hydrological events in 2016 exceeded by far that of any other type of natural hazards (Guha-Sapir and CRED, 2016). Concerning inundations, flooding was the third major cause of economic loss worldwide among all natural disasters between 2006 and 2015 (the firsts were earthquakes and storms), resulting in total damages larger then $ 300 billion. In Europe, the proportion of flood impacts was even larger during the same decade, with inundations ranked first in terms of total damage (i.e. $\sim$ $ 51 billion; CRED). The CRED findings about the increasing amount of economic loss starting from the second half of





$20^{th}$ century agree with the analyses carried out by the Intergovernmental Panel on Climate Change (IPCC), which highlighted that flood damages in the past ten years were ten times higher than in the period 1960-1970 (IPCC, 2001, 2014).

Future scenarios provided by IPCC (2014) and Jongman et al. (2012) suggest that extreme flood events at a global scale are expected to increase in terms of frequency and magnitude. Barredo (2009) drew an hypothetical scenario without any change

in the meteorological forcing and found that loss would increase anyway in the future due to exposure and socio-economic changes (e.g. higher demographic pressure, improved pro-capita wealth and living standards). According to Kvocka et al. (2016) and references therein, by 2050 66% of the population in the world will be living in urban areas and 40% of them will be located in flood-prone areas with high frequency of flood events. Therefore, the number of people potentially affected by floods (and consequently the amount of economic loss due to inundations) is expected to significantly increase in the near

future.

The implementation of the European Flood Directive (2007/60/EC) led flood risk assessment and management to gain even greater interest (de Moel et al., 2015; Dottori et al., 2016b, and references therein), forcing member states and authorities to dedicate additional resources and efforts to the assessment, mitigation and management of flood risk in the broader contexts of possible climate change, population growth and economic changes (Meyer et al., 2013; Merz et al., 2010, 2014). However,

despite these efforts, there are still several open problems and limits that need to be discussed and addressed in order to better assess flood risk and its evolution in time and space.

From an analytic point of view, flood risk is the combination of *hazard* (i.e. the probability of a flood event with a certain intensity to occur in a specific area and in a specific time period) and *consequences*, providing for instance information on the *vulnerability*, i.e. the type and number of elements affected by a given flood event, and how well they are able to resist.

According to one of the definitions proposed in the literature (see e.g., Merz et al., 2007), the *vulnerability* is a function of *exposure*, which indicates the quantification and qualification of the elements at risk, and flood *susceptibility*, namely the attribution of a loss value to the exposed element, as a function of one or more flood intensity parameters and resistance characteristics (damage models).

Uncertainty exists in all flood risk components (i.e., *hazard*, *exposure*, *susceptibility*, etc.) and, according to Merz and

Thieken (2009), it is appropriate to distinguish between epistemic and aleatory uncertainty. The first one refers to the difficulty to describe in detail the system in its entirety and in detail, because of scarce knowledge, generalizations, simplifying assumptions and aggregation of information. For example, epistemic uncertainty in hydrological and hydraulic modeling is associated with the necessarily simplified definition and simulation of hazardous scenarios; a simplistic schematization is also adopted to assess the elements at risk, which are often represented by coarse land-use maps. These generalizations introduce large sources

of uncertainty in the identification of the value of the elements at risk. In addition, we should take into account the aleatory uncertainty, which is due to the variability in space and time of the quantities that we consider in the analysis (e.g. market fluctuations, as far as the elements at risk are concerned; see de Moel and Aerts, 2011).

The scientific literature of the last decade shows a large number of innovative damage models that are capable of estimating flood loss starting from one or more predictive variables. Nevertheless, several authors indicate that damage models still provide

an important sources of uncertainty in flood damage estimates, leading to uncertainty which are comparable or larger to those




associated with any other component (Jongman et al., 2012; de Moel et al., 2012; Gerl et al., 2016; de Moel et al., 2014; Merz et al., 2004, 2007; Apel et al., 2009).

One important source of uncertainty is the simplified representation of complex damaging processes in terms of a stage-damage function (Jongman et al., 2012). Since White (1945) linked the water level to relative (i.e., the loss ratio) or total (i.e., in monetary values) damage, most of the models used today stick to this concept using only water depth to estimate relative loss (see e.g. Penning-Rowsell et al., 2005; Smith, 1994; Apel et al., 2009; Kreibich et al., 2009; Merz et al., 2013). Other important influencing factors, such as flood duration and flow velocity are often not considered (de Moel and Aerts, 2011; Merz et al., 2013).

Recently, some authors (see Merz et al., 2013; Chinh et al., 2016; Hasanzadeh Nafari et al., 2016, 2017; Kreibich et al., 2017; Spekkers et al., 2014) developed multi-parameter damage models including more than one predictive variable, chosen among other hydraulic parameters (e.g. streamflow velocity, duration of the inundation, etc.), resistance performance, precautionary measures and people awareness and experience with floods (Meyer et al., 2013). These models were shown to outperform uni-variable loss models, under the condition that sufficiently large and detailed damage data-sets are provided (Merz et al., 2013; Schröter et al., 2016). Bubeck and Kreibich (2011), Cammerer et al. (2013), Messner et al. (2007) and Meyer et al. (2013), among others, indicate the need for a better understanding of the damage processes as a means to further improve multi-variable models.

A further aspect that contributes to the uncertainty is the lack of sufficient, comparable and reliable high quality flood loss data (Meyer et al., 2013; Molinari et al., 2014a; Amadio et al., 2016; Scorzini and Frank, 2015; Green et al., 2011). In the absence of empirical damage data, damage models are either selected from the literature or subjectively and schematically derived by experts using a synthetic approach (see e.g. Penning-Rowsell et al., 2005; Merz et al., 2004; Thieken et al., 2008; Kreibich et al., 2010; Merz et al., 2013; Dottori et al., 2016a). In fact, data collected in the events aftermath are crucial to construct new models and validate existing ones (Meyer et al., 2013; Cammerer et al., 2013; Ballio et al., 2015), to adjust them for peculiar conditions of the study area, to improve the consistency of the models themselves (Amadio et al., 2016; Büchele et al., 2006; Gerl et al., 2016), and to provide information about their transferability in different analyses and contexts (Molinari et al., 2014a; Cammerer et al., 2013; Green et al., 2011). Many damage models developed up to now are in fact internationally accepted as standard methodologies of estimating flood damages (Merz et al., 2007; Smith, 1994; Merz et al., 2010), without being neither tested nor calibrated for the specific study area (Amadio et al., 2016). Indeed, using damage models for geographical areas, socio-economic conditions and flood events that differ from those for which the models themselves have been originally derived leads to the incorporation of large errors into the assessment of flood risk (Merz et al., 2004; Schröter et al., 2016; Merz et al., 2010). According to Gerl et al. (2016), validation analyses were performed only for about 45% of the existing literature models by means of comparisons with observed data, while for the remaining models either the evaluation status is unknown, or the validation process is not explicitly described.

Concerning Italy, the scientific literature reports on the one hand several examples in which models developed elsewhere are applied without calibration or validation (see e.g. Amadio et al., 2016), and on the other hand it clearly states the limited exportability of empirical damage models (see e.g. Molinari et al., 2014b, on the transferability of the model developed on the



basis of specific flood event data by Luino et al. (2006) and Freni et al. (2010)). Molinari et al. (2012) associate the generalized poor performance of loss models with a variety of reasons, among which two are worth recalling. First, the Italian peninsula is characterized by an extreme variability of geographical and geomorphological contexts as well as of urban patterns and building typologies. Second, Italian flood-loss data sets are generally of low quality and very often characteristic of small

areas, if compared to other European case studies (see Molinari et al., 2012).

## 1.1   AIMS AND STRUCTURE OF THE STUDY

We consider one of the most comprehensive flood damage data set in Italy, which consists of 1330 post-event data about flooded private properties, collected in the aftermath of the Secchia river inundation in the province of Modena (Northern Italy). The database contains information about the affected properties, such as their location and structural characteristics and

the amount of loss suffered, concerning both structural and non-structural parts and installations (termed "buildings" from here on) and furniture and household appliances ("contents") of each building (see Sec. 3.1 and 3.2). The raw data collected by local authorities has been homogenized, geocoded and integrated with other useful information including the outcomes of a hydronumeric simulation of the inundation event (see Sec. 3.3).

This study is structured into three main components:

- First, concerning direct tangible economic damages to buildings, we use the above data set to derive uni- and multi-variable damage models for the study area and compare the accuracy in estimating damages with a selection of established literature models.

- Second, we calibrate empirical uni- and multi-variable models to subsections of the study area and validate them using the data observed in different subsections (split-sample validation).

- Third, we investigate the relationship between damages to buildings and damages to contents, looking for the possibility to develop an empirical damage model also for the latter.

With this analysis, we contribute to the understanding of possibilities and limitations of flood damage modeling in Northern Italy with a particular focus on addressing the problem of lacking consistent data and the consequent difficulty in the development of reliable damage models for local applications. Also, our study investigates the open problem of transferability of

empirical damage models to different areas and socio-economic contexts. Finally, the analysis aims to provide further insight on accuracy and robustness of uni- and multi-variable models in estimating flood losses to buildings and content.

## 2   STUDY AREA AND INUNDATION EVENT

Our study focuses on a real inundation event occurred in Italy in 2014 and caused by a breach in the right embankment of the Secchia river during an intense, yet not extreme, flood event. The collapse of the right levee occurred on $19^{th}$ January near

the town of San Matteo, in the Northern part of the Modena municipality (see yellow dot in Fig. 1), and caused inundation of the neighbouring municipalities of Bastiglia, Bomporto and Modena (violet, orange and green polygons in Fig. 1, respectively)



in less than 30 hours. The overflowing volume was estimated between $36.3 \cdot 10^6$ and $38.7 \cdot 10^6$ m$^3$, flooding an area of about 52 km$^2$ (see e.g. Orlandini et al., 2015). Towns and surrounding countryside remained flooded for more than 48 hours, until a water volume in excess of 20 million cubic meters was finally pumped out of the inundated area. According to Orlandini et al. (2015), the total estimated flood loss was about € 500 million (about € 16 million considering only residential properties).

The study area includes the municipalities of Bomporto and Bastiglia and the Northern part of the Municipality of Modena. It is located on the downriver right side and it extends for approximately 112 km$^2$. The area is mainly flat and main relieves consist of roads or railways embankments and minor river levees. The aspect of the area is oriented in a North-Eastern direction, along which ground elevations decrease from ca. 30 m a.s.l. in the South-Western territories to ca. 18 m a.s.l., about 20 km North-Eastwards.

The delineation of the study area relies on different topographic boundaries. The Western boundary in Figure 1 is the right levee of the Secchia river, while the Eastern boundary consists of the left levee of the Panaro river, which also flows towards North-East, almost parallel to the Secchia river. Roads, embankments and drainage channels which form the Northern boundary are an important control for flooding dynamics (Carisi et al., 2017) and prevented urban areas further North from being flooded.

The breach was first detected at 6:30 a.m. Most likely it was triggered either by direct river inflow into the riverside entrance
of an animal burrow system or by the collapse of an existing animal burrow, which was separated by a 1 m earthen wall from the levee riverside and saturated during the flood event (Orlandini et al., 2015). A trapezoidal part of the embankment, with a base width of about 10 m, was removed and the embankment's top elevation became immediately 1 m lower than the river water surface. The breach reached a maximum bottom width of about 80 m and the embankment's top elevation became equal to the ground level within 9 hours (3:00 p.m. of 19$^{th}$ January 2014). Given the advanced state of the development of the breach
when it was first discovered, no repair of the breached levee was even attempted as immediate measure.

Thanks to several eyewitness accounts, video footage and studies conducted by the scientific committee (D'Alpaos et al., 2014; DICAM-PCREM, 2015), it was possible to identify the flood event propagation dynamics, shown by the blue arrows in Fig. 1. This data was used, together with local accounts, pictures and videos of the flooded municipalities, to reconstruct the event by means of a fully-2D hydrodynamic model (see Sec. 3.3).

**3  FLOOD LOSSES AND HYDRODYNAMIC DATA**

In the immediate post-event period, for the purpose of compensation, authorities of Emilia-Romagna Region, Modena Province and affected municipalities started a data collection campaign to get as much information as possible on the damages caused by the flood event. According to Regional Decree n. 8 of 24$^{th}$ January 2014, the aim of the survey was to quantify the financial needs for the restoration of damaged public buildings, infrastructure network, hydraulic and hydrogeological works,
as well as private properties for residential use, household contents, private registered goods and goods related to the productive sector. Accordingly, citizens and property owners were asked to fill forms about public properties damages (Form A), private properties, furniture and registered goods damages (Form B), economic and productive activities damages (Form C)





and agriculture and agro-industrial sector damages (Form D). In the present analysis, damage assessment focuses exclusively on private properties (Form B).

Authorities collected a total of 2448 forms, divided as per the affected municipalities. In order to geocode the position of every damaged property, the complete database was filtered, considering only records for which the complete address was

provided. The database regarded private properties affected by different kinds of potential damages: damages to buildings (structural and non-structural parts and installations), contents damages (furniture and household appliances), structural damages to common parts and registered goods damages (such as cars, motorcycles, etc.). Our analyses focused only on properties affected at least by damages to buildings. The total amount of considered forms is therefore 1330 (see Table 1, second column).

The 1330 records were geocoded in a GIS environment; geocoding was followed by a careful manual control activity using

publicly available internet pictures, Google Street View and Google Earth. This step enabled the correction of several wrong or inaccurate geocodings, mainly in the rural areas, where distances between street numbers are higher.

The refund requests by citizens, collected from municipal authorities, were divided into different asset typologies: buildings damages, contents damages, structural damages to common parts and registered goods. We neglected structural loss to common parts and registered goods in our analyses because of the limited amount of data collected on these categories. Table 2 shows in

details the different assets which could be refunded for buildings and contents damages. Table 3 summarizes all data collected and used in our study for each damaged property, providing information about the original sources and grouping the data into three different categories: observed (i.e. declared by owners in the official forms); simulated by the hydrodynamic model; retrieved from an external source. The last column of the same table reports the ranges of these variables within the study area. The following sub-sections detail the information collected and summarized in Table 3.

## 3.1  DAMAGES TO BUILDINGS

As mentioned before, all 1330 considered records reported damages to buildings (structural and non-structural parts and installations). Concerning this type of damages, authorities verified the authenticity of the owners declarations (who asked for compensations without knowing the refund criteria, just estimating the amount of the restoration work of the damaged parts) by means of experts evaluation in case of damages higher than €15 000 and defined the final compensation granted to owners

in accordance to Ordinance No. 2 of $5^{th}$ June 2014 and Law No. 93 of $26^{th}$ June 2014, which specifies the refund criteria. For instance, considering the total amount of money that authorities had available for the restoration of all kind of properties, the maximum coverage for each damage to buildings was set to €85 000, while each owner could receive up to €15 000 for contents damages, divided as follows:

- up to €5000 for the kitchen or, alternatively, €6000 for the living room with kitchenette;

- up to €2000 for each room and the living room (for a maximum of 3 refundable rooms);

- up to €1000 for each bath (for a maximum of 2 refundable baths);

- up to €2000 for a maximum of 1 appliance (e.g. garage, cellar, laundry).





It is understandable, therefore, that the limited availability of money and the need to find an objective criterion for all the affected properties led in many cases to the reduction of the amount of damages refundable to the owners. In fact, the refundable assets are only a percentage of the assets that can be found in a property and, in addition, the experienced damages could be higher than the maximum coverage established by authorities. The difference, in terms of total absolute buildings

damages, between refunded and claimed damages is equal to about € 2.1 million (€ 16.3 million of declared buildings loss vs. € 14.1 million of refunded buildings loss). Given these significant differences, in order to preserve the representativeness and consistency in loss data, we chose to consider the damages as claimed by citizens in the Form B (estimation of the financial need for restoration, without knowing the refund criteria) as observed loss in our study and all the analyses that will be illustrated in the reminder. We are aware that this choice can introduce overestimation of the damages, but we considered this eventual error

having less influence on loss estimation, both quantitatively and methodologically, with respect to the distortions that would be introduced systematically adopting the results of the compensation phase.

For the finality of the analysis, together with the amount of money requested for compensation, we extracted from the filled forms also the available information on building footprint and structural typology (masonry, reinforced concrete, etc.) because of their potential impact on the damage process and therefore on damage modeling (see also previous studies, e.g. Merz et al.,

15 2013).

In order to have the possibility to evaluate losses in relatives terms (as the percentage of damage suffered with respect to the total value of the building), we also retrieved the economic value of each property by means of the economic estimate provided by the Italian Revenue Agency (Agenzia delle Entrate - AE). Every six months AE issues the open-market values [€/m$^2$] for different assets (e.g. civil houses, offices, stores, etc.) in each Italian administrative district (spatial scale of municipality),

taking into account different classes of residential and industrial buildings and the overall economic well-being of the region. These values are different for each homogeneous geographical area (*OMI zone*) and set a minimum and a maximum market value per unit area. Focusing on residential buildings, we defined the building's economic values [€/m$^2$] as the average of the values provided for each property in the same *OMI zone*. It is important to notice that these values do not consider possible fall in price due to catastrophic events. Due to the absence of more specific data, the choice of this information at an aggregation

level seems to provide a sensible estimation of the economic value of properties, which are only partially damaged by floods and is in line with previous loss analyses at different scales (see e.g. Arrighi et al., 2013; Domeneghetti et al., 2015).

## 3.2 DAMAGES TO CONTENTS

We also analyzed in this study the monetary loss to household un-registered contents (e.g. furniture and household appliances: refrigerator, dishwasher, oven, sink, stove, washer, dryer, TV and personal computers).

Focusing on these data and looking at the refunded loss, because of the stricter criteria for contents damages compensation of Ordinance No. 2 of 5$^{th}$ June 2014 and Law No. 93 of 26$^{th}$ June 2014, this difference between requested and refunded amount is even more evident. It is equal to about € 6 million (€ 11 million of total declared loss to contents vs. € 5 million of total refunded contents loss) and confirms the choice to consider the damages as claimed by owners in the Form B as observed contents loss.





Concerning this data set, it is worth noting that we did not have any specific information for each building on the items recorded under the generic expression "contents". Therefore, we could not express these damages in terms of relative loss over the total movable property value. Also, the damage models to household content proposed by the scientific literature are fairly rare and isolated (some examples are represented by studies performed by Penning-Rowsell et al., 2010; Thieken et al., 2008).

Thus, we investigate the usefulness of an indirect modeling approach for this type of damages which is based on regressing losses to building content against losses to buildings.

## 3.3 HYDRODYNAMIC CHARACTERIZATION OF THE INUNDATION EVENT

Forms B collected from authorities for the purpose of compensation do not include data on hydraulic variables, such as water depth, water velocity, etc. Being these data necessary for the aim of our analysis, the reconstruction of the flood event was

performed by means of a 2D finite element numerical model (Telemac-2D) a fully-2D hydrodynamic model which solves the 2D shallow water Saint Venant equations using the finite-element method within a computational mesh of triangular elements (see Galland et al., 1991; Hervouet and Bates, 2000, for details). This computational model complies with the validation protocol by the International Association of Hydraulics Research (IAHR) and has been successfully applied to case studies around the globe (Hervouet and Bates, 2000; Brière et al., 2007).

Concerning the inundation event, the dynamics of the wetting front was strongly influenced by the presence of topographic discontinuities (e.g. road embankments, artificial as well as natural channels belonging to the minor stream network, etc; see D'Alpaos et al., 2014). In order to correctly reproduce the ground elevation and the discontinuities in the model, a detailed LiDAR DEM with spatial resolution of 1 m was used and an unstructured triangular finite element mesh of the study area was generated. The mesh consists of $34\,082$ nodes connecting $66\,596$ elements with variable size from 1 to 200 m in the flatter

zones, covering a total of 112 $km^2$. This accurate mesh ensures the correct representation of all major linear discontinuities existing in the study area.

The outflowing hydrograph of the levee breach as reconstructed by the scientific committee that studied the event (D'Alpaos et al., 2014) was used as boundary condition, in particular as inflow to the boundary elements representing the levee breach.

The calibration of the 2D model was performed by varying the floodplain roughness coefficients in order to reproduce the

real extent of the inundation, at different time steps, as documented by maps and aerial images made available in the immediate post-event by competent authorities and rescuers (D'Alpaos et al., 2014), and as also confirmed by later studies (see e.g. Vacondio et al., 2016). In particular, the Manning's coefficients values were differentiated between agricultural areas and urban areas, and the resulting coefficients (0.033 $m^{-1/3}$s and 0.1 $m^{-1/3}$s, respectively) are in line with the values reported in the scientific literature (see e.g. Vorogushyn, 2008; Domeneghetti et al., 2013).

After the event, local authorities collected information about the water depth reached in different points of the inundated area. This information was used for the validation of the model, together with pictures, videos and reports made available on the Internet sites, as well as in situ interviews. In about 50 points, uniformly distributed in the study area, simulation outcomes were compared in terms of water depth with the information available. Results showed a good agreement between simulated





and observed flooding dynamics, being the residuals between observed and simulated water levels always smaller that $\pm 20$ cm.

The calibrated and validated model was then used to reconstruct the detailed spatio-temporal dynamics of the inundation event and to identify the spatial distribution of the hydraulic variables of interest. In fact, combining the 2D model outcomes and the geocoded locations shown in Fig. 2, it was possible to extract at each point the maximum water depth, the maximum flow velocity and the duration of the inundation (see Table 3).

## 4 DAMAGE MODELS

As already discussed in Sec. 1, damage models return the amount of loss potentially suffered by certain elements (population, buildings, economic activities, ecosystem, etc.) as a result of a specific flood event, thus providing an estimate of the object's susceptibility. These models associate relative (or absolute) losses with different input variables. The most frequently used models in Europe are uni-variable damage models, i.e. they estimate the amount of relative damages as a function of a single input variable, most commonly water depth, (Merz et al., 2010; Messner et al., 2007; Jongman et al., 2012), differentiated by building use, type, etc. (Gerl et al., 2016).

This section briefly recalls well known and largely employed literature depth-damage models (also called "stage-damage models", shown in Fig. 3), as well as two empirical depth-damage models and one multi-variable loss model that we identified for the Secchia loss data set . All uni- and multi-variable models illustrated here are applied for predicting loss to household contents resulted from the January 2014 Secchia flood event.

### 4.1 LITERATURE DAMAGE MODELS

#### 4.1.1 Multi-Colored Manual (MCM)

The damage curve implemented in the Multi-Colored Manual (MCM; Penning-Rowsell et al., 2005) is considered as one of the most comprehensive and detailed models for flood damage estimation in Europe and is used as a support for water management policy and quantitative assessment of the effect of investment decisions (Penning-Rowsell et al., 2010; Jongman et al., 2012). It estimates different kinds of expected loss (e.g. loss to building structure, equipment, immobile inventory, mobile inventory, stock; see Kreibich et al., 2010) as a function of the local water depth, like other stage-damage functions. Differently from the majority of other damage models, the MCM model estimates buildings damages using absolute depth-damage curves, i.e. it defines monetary potential loss related to water depth, rather than providing damages percentage (Penning-Rowsell et al., 2005; Bubeck and Kreibich, 2011; Jongman et al., 2012). This stage-damage model estimates loss for a wide variety of residential, commercial and industrial buildings, based almost exclusively on synthetic analysis and expert judgment from the insurance industry or engineers, and it evaluates the amount of damages that would occur to a specific element at risk under certain flood conditions (Penning-Rowsell et al., 2005; Bubeck and Kreibich, 2011). Aiming at performing a fair comparison





between all considered models, instead of the absolute depth-damage curve we considered a MCM relative curve, obtained referring to the average economic value of the buildings of the Secchia study area.

### 4.1.2 Flood Loss Estimation MOdel for private sector (FLEMOps)

The "Flood Loss Estimation MOdel for private sector (FLEMOps)" (Thieken et al., 2008) is an empirical model based

on an extensive data set from 2158 private households that were significantly affected by flood events in 2002, 2005 and 2006 in Germany. According to Thieken et al. (2008), the database used for identifying FLEMOps was compiled through computer aided telephone interviews with a sample of people affected by these serious events. The interviews consisted of 180 questions conceived to reconstruct the flood details, that is the main hydraulics features and the type of damage suffered by the households. The FLEMOps model assesses relative flood damages for private households referring to several factors:

- inundation depth ($h$) (five classes: $h \leq 20$ cm, $h = 21\text{-}60$ cm, $h = 61\text{-}100$ cm, $h = 101\text{-}150$ cm, $h > 150$ cm);

- building types (three classes, in order to capture differences between the value of different buildings: family-houses, (semi-)detached houses, multifamily houses);

- building quality (two classes, representative of the state of conservation of a building, that can influence both the economic asset value and the building resistance to the water flow: low/medium quality, high quality);

- water contamination (three classes in order to represent the degree of pollution of the inundation water: none, medium, heavy (i.e. oil or multiple) contamination);

- private precaution (three classes reporting the existence of preventive measures for flood risk mitigation: none, good, very good precaution).

Although the original FLEMOps model has been developed as a multi-variable model, in this study we implemented it as

a uni-variable one, referring to the water depth as the only parameter available in our data collection. The curve taken into account in this study is the one that considers a uniform distribution of building types in the study area (see Apel et al., 2009), while no information about building quality, water contamination and private precaution were available (concerning these last three factors, the first classes of the original model were considered).

### 4.1.3 Rhine Atlas damage model

The "Rhine Atlas damage model" was designed for the hydraulic risk assessment within the watershed of the Rhine river, where to date, over 10 million people live in area with a very high flood risk. In 1993 and in 1995 two severe floods caused a large amount of economic damage in Germany and the evacuation of 250 000 people in the Netherlands (Bubeck et al., 2011). After these floods, in 1998 the International Commission for the Protection of the Rhine (ICPR) worked to identify and reduce flood risk in the Rhine river basin (Jongman et al., 2012) and in 2001 developed the Rhine Atlas damage model,

in which the damage intensity and the maximum damage values were established on the basis of the collected empirical data in the two mentioned floods and experts judgements, combined with a synthetic approach (Bubeck and Kreibich, 2011). This



model includes five different stage-damage functions, each of which is associated with a different land-use class derived from CORINE Land Cover project (European Environment Agency, 2007). Figure 3 shows the Rhine Atlas damage model used in this analysis, i.e. the stage-damage curve associated with the residential sector.

### 4.1.4 Joint Research Centre (JRCs) damage models

These curves were developed by the European Commission's Joint Research Centre - Institute for Environment and Sustainability (JRC-IES) (Huizinga, 2007) as part of a project to estimate trends in European flood risk under climate change (Ciscar et al., 2011; Feyen et al., 2012). These curves consist of different depth-damage functions and maximum damage values which can be used by all EU countries. On the basis of the land-use data retrieved from the CORINE project (European Environment Agency, 2007), five damage classes were established: residential, commercial, industrial, roads and agriculture. Stage-damage

functions were identified for ten countries from existing studies (for example, depth-damage models based on Penning-Rowsell et al. (2005) and van der Sande (2001) were used to develop a stage-damage model for the United Kingdom and, regarding Germany, depth-damage functions were chosen using a combination of many existing models; see Jongman et al., 2012) and applied to the corresponding damage classes. In addition, an average of all available land-use specific curves was used to develop a model for the countries, where stage-damage curves were not available ("JRC other countries" model), and Italy is

among these (Manciola et al., 2003; Molinari et al., 2012). We selected for our analysis seven out of the eleven JRC available curves: we neglected the curves that provide the highest and the lowest damage estimation for water depths between 0 and 2.5 m, that is the range that includes our observed data. In fact, these curves would be located respectively above and below the observed grey data points in Fig. 3, and would provide unrealistic over- and underestimations for our case study. Therefore, the curves that we considered for our analysis are: JRC Belgium, JRC Czech Republic, JRC Germany, JRC Netherlands, JRC

Switzerland, JRC UK and JRC other countries.

### 4.2  MODELS DEVELOPED ON SECCHIA DATA SET

### 4.2.1  Secchia Empirical (SEMP) damage model

    The "Secchia Empirical (SEMP) damage model" is an empirical stage-damage curve that we derived from the observed relative loss for the inundation event of 2014. It was obtained by binning water depth values into classes of 25 cm each (i.e.

0-25cm; 25-50cm; etc.) and by calculating the median damage for each bin. Then, for each bin the median damage value was associated with the mean water depth of the bin itself (e.g. 12.5 cm; 37.5 cm; etc.), and the empirical damage curve was then obtained by linear interpolating the binned values. This curve is obviously limited to the maximum water depth observed in the 2D simulation. Different classes subdivisions were tested (from 10cm to 1 m water depth) and the one chosen (25 cm) resulted to be the one with the best performance in reproducing observed loss data.





### 4.2.2 Secchia Square Root Regression (SREGₓ) damage models

We obtained the "Secchia Square Root Regression (SREG) damage models" by regressing observed relative loss against: maximum water depth (SREG$_d$); maximum water velocity (SREG$_v$); and building footprint or area (SREG$_a$) recorded for every buildings, respectively. It is worth pointing out that SREG$_a$ refers only to footprints of buildings that are flooded during the

considered event (i.e. a real inundation or a flooding scenario). We tested linear, logarithmic and square root regression of the observed data, obtaining the best prediction performance in terms of Root Mean Square Error (RMSE) with the latter.

The identified regression relationships read:

$$D_{SREG_d} = 0.052\sqrt{h} + 0.059 \tag{1}$$

$$D_{SREG_v} = 0.027\sqrt{v} + 0.093 \tag{2}$$

$$D_{SREG_a} = -0.003\sqrt{a} + 0.135 \tag{3}$$

where $D_{SREG_d}$ [-], $D_{SREG_v}$ [-] and $D_{SREG_a}$ [-] represents relative economic damages to buildings estimated referring to the maximum water depth $h$ [m], maximum water velocity $v$ [m/s] and building area $a$ [m$^2$], respectively.

### 4.2.3 Secchia Multi-Variable (SMV) damage model

The "Secchia Multi-Variable (SMV) model" of this study took advantage of the Secchia 2014 data set by applying a similar

procedure to the one used to develop and validate an existing model at the German Research Centre for Geosciences (GFZ) (see Merz et al., 2013). While the approach used by Merz et al. (2013) was based on Bagging Decision Trees, using the corresponding Matlab toolbox implementation, the multi-variable model presented here used the Random Forest methodology, based on the R package randomForest.

Similarly to the Bagging Decision Trees one (Merz et al., 2013), the model consists of many regression trees, which are

tree-building algorithms for predicting continuous dependent variables. The procedure of growing each tree consists of the approximation of a non-linear regression structure, recursively repeating a sub-division of the given data set into smaller parts, in order to maximize the predictive accuracy of the model. The classification and regression tree (CART) methodology (Breiman et al., 1984) is used to select and split variables (splitting criterion) and to identify leaf nodes (stop criterion). It uses an exhaustive search method on a randomly chosen set of variables to identify the variable with the best split based on a

measure of node impurity (in our case the RMSE of the response values in the respective parts). The splitting is stopped either if a threshold for minimum number of datapoints in leaf nodes is reached or if no further splitting is possible. These steps create a tree structure with several nodes, whereby the beginning node is called root node and the last nodes are called leaf nodes





and each resulting node of the tree represents the answer to the partition question asked in the previous interior nodes. The prediction for an input $x_1$, $x_2$, ..., $x_k$ depends of the response variable of all the parts of the original data set that are needed to reach the terminal node (Merz et al., 2013). A possible problem of regression trees is overfitting, i.e. growing trees that are too large and with many leaves some of which are associated with small subsamples. The consequence is that the model works well

with the training data but have a large uncertainty on the validation with independent data. In order to reduce the uncertainty associated with the selection of a single tree, Breiman (2001) proposed the so-called Random Forest (RF) algorithm, in which multiple data set subsamples are created using the resampling bootstrap method and classification and regression trees are then developed for each bootstrap sample, considering a limited number of variables at each split to learn the trees. All the trees are then evaluated together and as reliable response the value is chosen, which represents the average of the responses from the

individual regression trees.

The RF algorithm has the advantage of providing estimates regarding the importance of variables in the tree-building process, and thus, in our case, of evaluating the relative importance of the contribution of each independent variable in representing the damage process: randomly permuting the values of the predictor variables, the algorithm simulates the absence of a particular variable and calculates the difference of the prediction error with and without the permutation. The variables being randomly

permuted presenting a low accuracy are the most important ones in the damage prediction, as their influence in the prediction process is very high.

The RF algorithm was used in many different scientific fields, from flood hazard assessment (Wang et al., 2015) to computer-aided diagnosis (Mihailescu et al., 2013), passing through gene selection (Deng and Runge, 2013), earthquake-induced damage classification (Solomon and Liu, 2010) and many others. The numerous applications show the many advantages of using the

RF method, including high prediction accuracy, acceptable tolerance to outliers and noise, and easy avoidance of overfitting problems. In the last years, some applications of this method to flood risk have been performed (see Merz et al., 2013; Chinh et al., 2016; Hasanzadeh Nafari et al., 2016, 2017; Kreibich et al., 2017; Spekkers et al., 2014), but literature in this field is still scarce if compared to the numerous studies that use simpler uni-variable models. Nevertheless, Merz et al. (2013) demonstrated that tree based models are able to improve the performance of existing models like stage-damage functions and

to better identify the most informative independent variables and their interactions (e.g., they can identify different importance levels of a same variable, depending on the value of another variable). Another important advantage of this learning machine is the possibility to include both continuous, e.g. water depth or velocity, and categorical variables, e.g. building type. On the other hand, these kind of multi-variable models are associated with some disadvantages: the most affecting one is the large amount of data needed in order to correctly identify complex relationships between variables, especially in geographically large

areas. This is one of the reasons why this kind of models is scarcely used in regions where comprehensive, multi-dimensional databases are not available (Merz et al., 2013).

We considered in our model all the variables that were available, collected from authorities, simulated by means of the hydrodynamic models and retrieved from external sources: maximum water depth, maximum water velocity, flood duration, buildings area, economic buildings value and structural typology.





## 5 RESULTS AND DISCUSSION

### 5.1 LITERATURE AND EMPIRICAL DAMAGE MODELS COMPARISON

Figure 5 shows the results of an analysis of the correlation between the relative flood loss to buildings and six predictive variables: maximum water depth, maximum water velocity, flood duration, building value, building area and structural typol-
ogy. Being the latter a categorical variable, it was converted to dummy variable encoding in order to calculate the correlation of continuous and categorical data together. We referred to the Spearman correlation coefficient in order to take into account also non linear relationships between variables and ordinal variables.

Empty boxes represent correlation that are not statistically significant at a 5% significance level. The only variables that resulted significantly correlated with the relative loss to buildings were the maximum water depth, building value and structural
typology. However, correlations coefficients between these variables and relative damages are low, precisely lower than $\pm 0.18$. Pearson correlation was also calculated and the resulting coefficients were similar to the Spearman's correlations (not shown).

Figure 6 shows the output of the evaluation of the importance of the variables taken into account in the loss estimation, performed by the SMV model on the basis of the six used variables (building area and value, flood duration, maximum water velocity and water depth, structural typology). One of the advantages of this kind of multi-variable models, in fact, as discussed
in Sec. 4.2.3, is the possibility to understand the influence of the factors on the damage process for this specific context (different concept from the correlation one). In contrast to other studies, e.g. (see Merz et al., 2013) the data set does not reveal a distinct importance for individual variables, event not water depth does not stand out. The descriptive capability of water depth is only slightly stronger than water velocity and building area, while the remaining predictors show very small importance.

Figures 3, 7, 8 and 9 show in the background the observed relative damage to buildings, collected in three municipalities (i.e.
Bastiglia, Bomporto and Modena) as a function of maximum water depth (the first two figures), water velocity and building area, respectively. Despite the statistically significant correlation of water depth (see Fig. 5), a very large noise can be observed in the diagrams, which implies that one variable alone explains only a very limited part of the damage process. This is confirmed from the outcomes of both the correlation assessment and the importance analysis.

Taking the maximum water depth as only explanatory variable, beside the observed loss values Fig. 7 represents the damages
to buildings estimated by means of the uni-variable models developed on Secchia data set (SEMP, with blue dots, and SREG_d, dark red dots). With the same approach, Fig. 8 and 9 show the relative loss to buildings as function of maximum water velocity and building area, respectively, estimated by means of SREG$_v$ and SREG$_a$ models (dark red dots in both figures). Results of the application of the multi-variable model (SMV model), described in Sec. 4.2.3, are shown in Fig. 10, where relative damages to buildings estimated with the SMV model are compared with the observed loss.

The good performance of the multi-variable SMV model is already visible in Fig. 10, but it is shown more clearly in Table 4,
which reports the discrepancy between observed ($O_i$) and predicted ($P_i$) loss values with the local empirical models in terms





of three different performance metrics, namely BIAS, Mean Absolute Error (MAE) and Root Mean Square Error (RMSE), which are defined as follows:

$$BIAS = \frac{1}{n}\sum_{i=1}^{n}(P_i - O_i) \tag{4}$$

$$MAE = \frac{1}{n}\sum_{i=1}^{n}|P_i - O_i| \tag{5}$$

$$5 \quad RMSE = \sqrt{\frac{1}{n}\sum_{i=1}^{n}(P_i - O_i)^2} \tag{6}$$

SMV is associated with the lowest RMSE value (i.e. 0.062), which is the half of the RMSE value of the second best model (i.e. the $SREG_d$ model, with an RMSE value of 0.124). SREG models based on maximum water velocity ($SREG_v$) and building area ($SREG_a$) also provided relative loss estimation with almost identical results. RMSE referred to SEMP model is equal to 0.130. Results are similar in terms of BIAS and MAE, although some differences can be pointed out for the $SREG_x$ models, which present an BIAS value that is slightly lower than the one derived from the SMV model estimation.

Concerning literature models described in Sec. 4.1 and illustrated in Fig. 7, Table 5 shows that the best performance come from the FLEMOps and JRC Czech Republic models, which present values of RMSE, equal to 0.125 and 0.127, respectively. Although this values are satisfying in terms of errors, the performance of this models are lower than the ones of the models developed on Secchia's data set (except SEMP model). RMSE values derived from the relative loss estimation with JRC Netherland, JRC Germany, JRC Belgium and Rhine Atlas are between 0.13 and 0.15, while the worse performance in terms of RMSE resulted by JRC Switzerland, JRC other countries, MCM and JRC UK models. These outcomes reflect the fact that these latter damage curves are all in the upper part of Fig. 3, and significantly apart from the rest of the models, which are instead close to each other. Results in terms of BIAS and MAE reflected the ones analyzed before.

Analogous results can be observed in terms of absolute monetary loss in €, calculated as relative loss times the building values. The last column of both Table 4 and 5 reports the differences (in percentage) between the total observed absolute damages to buildings (€ 16.3 million) and the total absolute loss to buildings estimated by means of the study uni- and multi-variable models. SMV seems to have slightly worse performance than $SREG_d$, $SREG_v$ and $SREG_a$ (and FLEMOps, regarding these specific outcomes).

It is also worth noting that six out of fifteen tested models (considering literature and local models together) underestimated the total absolute loss, while the remaining nine models overestimated them. As far as what the literature damage models concerns, the loss overestimation with JRC UK, MCM, JRC other countries, JRC Switzerland and JRC Belgium models can be expected already observing Fig. 3, where the cited models are situated in the upper part of the graph, above the most of the observed damage points. The reason behind this fact must be attributed to the morphologic and socio-economic context where





this models have been drown, that differs considerably from the Secchia ones, in addition to the different criteria adopted to develop them.

Concerning the empirical models based on Secchia data set, the results reported in Table 4 referred to a calibration of the model using the entire data set. A study on the validation of all models was performed in addition, using instead separate
data sets for developing the model and for validating it. Specifically, one third of the records was randomly selected from the data set, and the model (calibrated on the remaining data) was applied on these records. BIAS, MAE and RMSE calculated in this context and reported in Table 6, showed values that are very similar to the ones reported in Table 4 concerning the $SREG_x$ and SEMP models. Results of the validation of the SMV model by means of the same approach, instead, indicated lower performance of this model, when calibrated on a smaller data set (see Table 6). In fact, values of BIAS, MAE and RSME
are twice as high as the values reported in Table 4, which refer to the calibration of the models on the entire database. These outcomes further highlight the need for extensive data sets to be able to identify robust and reliable damage models. From the comparison of the different models considered (uni- and multi-variable), it is clear that this aspect is more evident in the case of the multi-variable model, for which the performance in the damage estimation is significantly worse when calibrated on a smaller number of observed data. On the contrary, uni-variable models, though simpler than the SMV model, appear more
robust in case of a smaller amount of calibration data, providing better results in the validation.

## 5.2   VALIDATION OF LOCALLY DERIVED DAMAGE MODELS

Based on the output in Sec. 5.1, it is worth noting that the application to the Secchia case study of the JRC other countries model, in which Italy should be included, provided very poor results in terms of building loss. This confirms how challenging it is to identify a regional or large scale model with a general validity (see also Sec. 1 and Cammerer et al., 2013; Amadio et al.,
2016; Molinari et al., 2012).

This section further assesses the transferability of damage models calibrated against observed loss data to very similar socio-economic contexts. We developed $SREG_x$, SEMP and SMV models on the basis of the entire data set (a total of 1330 observed records in our case) and they showed a fair, or good, prediction performance for the entire study area. In order to test the transferability of such models to similar contexts, we identified analogous models ($SREG_x$, since it resulted to be the best
model among the local derived ones, and SMV models) on the basis of the loss data collected in a single municipality and then applied these models for predicting flood loss in a neighboring municipality, concerning damages to buildings. In particular, among the three municipalities considered in the study (i.e. Bomporto, Bastiglia and Modena), we neglected Modena due to its limited number of observed monetary loss (51 observed records), while we considered Bastiglia (887 observed records) and Bomporto (392 observed records). We then calibrated the Square Root Regression models on Bomporto' subset ($Bo\_SREG_d$,
$Bo\_SREG_v$ and $Bo\_SREG_a$) and on Bastiglia subset ($Ba\_SREG_d$, $Ba\_SREG_v$ and $Ba\_SREG_a$), and we applied Bomporto's and Bastiglia's Square Root Regression models for predicting Bastiglia and Bomporto flood damages to buildings, respectively. We finally performed a similar resampling experiment considering multi-variable models, identifying $Bo\_MV$ and $Ba\_MV$ models on the basis of Bomporto and Bastiglia subsets and using these models for predicting flood loss observed in Bastiglia and Bomporto, respectively.





Figure 11 shows the part of the results of these resampling experiments. The figure in the top panel refers to Bastiglia's relative damages to buildings, estimated via (Bo_SREG$_d$) model, while the bottom panel indicates Bomporto's damages estimated via (Ba_SREG$_d$) model; in each graph grey dots represent observed loss, red dots indicate relative damages to buildings estimated with Square Root Regression models, and finally blue dots show the estimation of relative loss using the MV models.

SREG$_x$ models shows rather poor performances, being capable of capturing the average loss only, while better performance seem to be associated with MV models in both graphs. It is worth noting some differences between the two panels: grey dots in the upper panel (application to Bastiglia of the models calibrated in Bomporto with 392 data) seem to overestimate the relative loss to buildings, while in the lower panel (application to Bomporto of the models calibrated in Bastiglia with 887 records) they lie closer to the bisector. The studies in terms of relative damages to buildings related to maximum water velocity and building area present very similar results, that are omitted for the sake of brevity.

This outcome is also visible in Table 7, which presents the results of the resampling experiments in terms of the usual indexes BIAS, MAE and RMSE.

While uni- and multi-variable models calibrated on Bastiglia's data and applied with Bomporto's subset of loss data do not differ much, with slightly better performances for the MV class of models, the multi-variable model derived from Bomporto's subset of data applied to Bastiglia's one is associated with much higher prediction errors. The same cannot be observed for SREG$_x$ models' results, which are all comparable to each other. The worse performance of the Bo_MV model applied to Bastiglia's subset of damage data can be explained by the smaller size of the Bomporto subset of data, which was used for identifying the model itself and is less than a half of the Bastiglia's sample. As outlined in Sec. 4.2.3, in order to have robust results from MV models, a large amount of empirical data is required. Furthermore, this study gives preliminary results to affirm the importance of having a sample size reflecting the extent of the area it refers to. Bastiglia flooded area is less than half the Bomporto's one (see Fig. 2), yet Bastiglia's sample is more than twice as big as Bomporto's one. This explains rather clearly the difference in terms of accuracy of the Ba_MV and Bo_MV models in Table 7, the higher the loss data density the better and more robust the representation of the relationship between different predictor variables and loss data and the higher the ability of the model to explain local characteristics of the study area (Schröter et al., 2014).

## 5.3 MODELING FLOOD LOSSES TO CONTENTS

As for the damages to buildings, first of all we analyzed the Spearman correlation between the observed flood loss to contents and all potential predictive variables (i.e. maximum water depth, maximum water velocity, flood duration, building value, structural typology, building footprint, or area, and absolute damages to buildings). Figure 12 shows the results of this assessment, where full boxes represent statistically significant correlation coefficient at a 5% significance level. On the one hand, similarly to the analysis for building losses, the maximum water depth and the structural typology resulted to be significantly correlated with damages to contents, although their correlations coefficients are low. On the other hand, damages to contents turned out to be significantly correlated with the building footprint (Spearman correlation coefficient equal to 0.27) instead of the building value. A noteworthy feature of Figure 12 is the very strong and statistically significant positive correlation between damages to buildings and to their content (Spearman correlation coefficient equal to 0.59).





We therefore explored in our study the possibility to exploit the relationship between monetary losses to buildings and content for predicting these latter. We tested different types of mathematical relationships (i.e. linear, square-root, logarithmic and bilogarithmic regressions), and the square-root regression resulted the one with the best prediction performance in terms of RMSE, i.e. the one that best relates monetary losses to buildings with those to contents. In fact, the RMSE coefficient is equal
to € 10 742, while it resulted to be € 11 159, € 11 184 and € 11 527 for linear, logarithmic and bilogarithmic relationships, respectively. The identified regression relationship reads:

$$D_{contents} = 125\sqrt{D_{buildings}} - 1966 \tag{7}$$

where $D_{contents}$ [€] represents economic damages to contents, while $D_{buildings}$ [€] indicates loss to buildings. Fig. 13 depicts empirical vs. predicted monetary loss to contents.

The last component of our analysis applied Equation 7 for estimating damages to contents using estimates of buildings monetary loss resulting from the uni- and multi-variable damage models that we considered in our study, instead of observed damages. Table 8 lists the performance metrics BIAS, MAE, RMSE obtained while predicting monetary loss to contents as described, as well as the relative difference (%) between empirical (i.e. € 11 million) and predicted total monetary loss to contents. The first row in Table 8 reports as a reference term the same performance indexes that can be obtained when Eq. 7 is
applied with observed damages to building.

The outcomes reflect the results that we obtained when modeling buildings losses, presented in Sec. 8. Evidently, models associated with poor performances in predicting monetary losses to buildings are also not reliable for indirectly predicting losses to building content (i.e. JRC Switzerland, JRC other countries, MCM and JRC UK). As reported in Table 8, the ranking of the best performing models in terms of RMSE for an indirect assessment of losses to content is JRC Netherlands (€ 12 702
), SEMP, JRC Germany, JRC Czech Republic, Rhine Atlas, SREG$_v$, FLEMOps, SREG$_a$, SREG$_d$, JRC Belgium and SMV (€ 15 292). The performance of all considered models, with the exception of the last four in Table 8, show a difference between observed and predicted overall monetary losses to contents that does not exceed € ±4 million (except for JRC Belgium that presents a difference value of € 7.2 million). JRC Netherlands, SEMP, JRC Germany, SMV and JRC Czech Republic are associated with differences lower than € ± 2 million. Unlike the results obtained when predicting damages to buildings, most
of damage models seemed to overestimate contents loss, while JRC Netherlands, SEMP, JRC Germany and Rhine Atlas slightly underestimated them.

## 6  Conclusions

Our study focuses on flood loss modeling for a comprehensive and extensive database of observed damage data (1330 records), which were collected after a recent inundation event in Italy. The event caused by a breach in the right embankment of
the Secchia river, in the Northern part of Modena's municipality. We derived empirical uni- and multi-variable damage models, whose performance has been compared with that of stage-damage functions existing in the literature (MCM, FLEMOps, Rhine Atlas and JRC models for different countries).




Consistently with the findings of Cammerer et al. (2013), Dottori et al. (2016a) and Scorzini and Frank (2015), locally identified empirical models provide better estimation of relative and absolute damages to buildings. This result underlines criticality and uncertainty associated with the application of literature damage models to different context from the ones in which they were originally developed.

5     Even though some literature models have similar performance to locally identified empirical models, the best performing literature models cannot be identified a-priori, which hampers the practical utilization of literature models themselves for predictive purposes.

    Concerning the estimation of relative loss to buildings, the Secchia Multi-Variable (SMV) model demonstrates slightly better performance (except for the differences between estimated and observed data) than other models. This outcome, however, is 10   not confirmed with regards to the contents damages.

    According to Elmer et al. (2010), Schröter et al. (2014) and Schröter et al. (2016), the use of a number of explanatory variables to sustain more complex models (i.e., multi-variable model) leads to additional knowledge of the event, especially if the interdependence of the parameters are considered. However, this may introduce additional uncertainties, especially if the additional parameters are not collected specifically aiming at this kind of analysis. As a matter of fact, Secchia's database 15   was collected for other purposes and does not include hydraulic parameters. Further uncertainties on the data set come from the records' geocoding (see Sec. 3), which may not match perfectly with the real location, thus influencing the assignment of the hydraulic parameters. Moreover, the building values provided by the Italian Revenue Agency (Agenzia delle Entrate - AE) represent the buildings market values at a given time of given building typologies, that is more an expression of the overall economic well-being of a specific area rather than the depreciated economic buildings values in case of a flood event. All these 20   sources of uncertainty may undermine the potential added values attributed to large flood damage data set.

    Although it did not seem to provide real important improvements in the estimation of flood loss in this case study, regression trees composing the multi-variable (MV) forest provide the important advantage to avoid the need to find a parametric function that works with all the data. Also, MV provide useful information about the relationship among the variables and how to exploit the local relevance of predictors. These can be very useful information for authorities and stakeholders to define preventive 25   measures and/or mitigation strategies.

    However, as the outcomes of the models transferability clearly highlighted and in order to lead to satisfying results, the use of this kind of multi-variable models requires a sufficient amount of data (Merz et al., 2013; Schröter et al., 2014). To completely exploit the potential of such models and sustain the possibility to export their use in different areas is necessary to pursue a detailed and structured acquisition of explanatory variables. According to Amadio et al. (2016), Molinari et al. (2012), 30   Molinari et al. (2014b), and Scorzini and Frank (2015), the most urgent need in Italy, as far as loss estimation is concerned, is to identify guidelines, valid for the whole country, to collect consistent and comparable data, even if they relate to different contexts. This data should include further useful information in addition to those commonly collected, such as e.g.: observed water depths; flood duration; presence of sediments; contamination rate; early warning or precautionary measures adopted; as well as other indication about the buildings composition (numbers of floors, type of contents, presence of basements, building 35   condition, etc.), preferably collected in the immediate post-event (see Merz et al., 2010).



As emerges from this analysis, in case of limited and uncertain information, the empirically uni-variable models derived in this case study still represent a good compromise between model complexity and reliable damages estimation results. Unlike other literature models developed for site-specific application and rarely tested for transferability, this study demonstrates that models can be transferred to similar contexts with satisfying results. Since the creation of a "one-size-fits-all" model is almost

5 impossible due to large variability of geographical and geomorphological contexts as well as urban patterns and building typologies in Italy, the definition of various damage models for different standardized Italian contexts is of large importance to increase the reliability of future flood risk analyses. The adoption of probabilistic modeling concepts could add another useful level of detail in terms of quantitative information about the uncertainty.

Finally, our study also emphasizes that loss-data collection is a fundamental and delicate task, and data-collection protocols

10 are urgently needed for harmonizing and standardizing the compilation of flood-loss data sets.

*Competing interests.* No competing interests are present

*Acknowledgements.* Emilia-Romagna Region, Regional Agency for Civil Protection, and Po River Basin Authority are kindly acknowledged for providing the data sets used in this study. In fact, part of the activity was performed with the support and contribution of the Civil Protection Agency of Emilia-Romagna under a five-year framework research agreement with the Department of Civil, Chemical, Environmental and

15 Materials Engineering (DICAM) of the University of Bologna (DICAM-PCREM, 2015). The present work was developed also within the framework of the Panta Rhei Research Initiative of the International Association of Hydrological Sciences (IAHS). Funding was partly provided by the University of Bologna, the SYSTEM-RISK Marie-Skłodowska-Curie European Training Network (EU grant 676027).





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





**Figures**

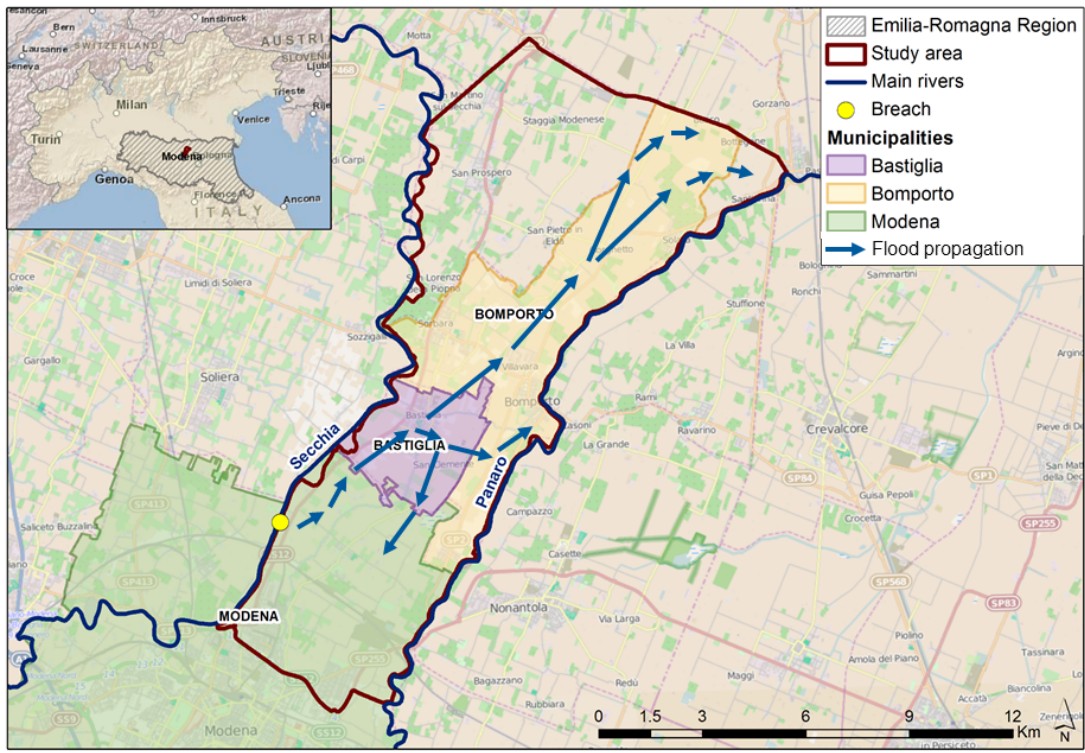

**Figure 1.** Study area: Secchia and Panaro rivers; location of the breach (yellow dot); municipalities of interest (i.e. Bastiglia, Bomporto and Modena); schematic of the inundation dynamics.




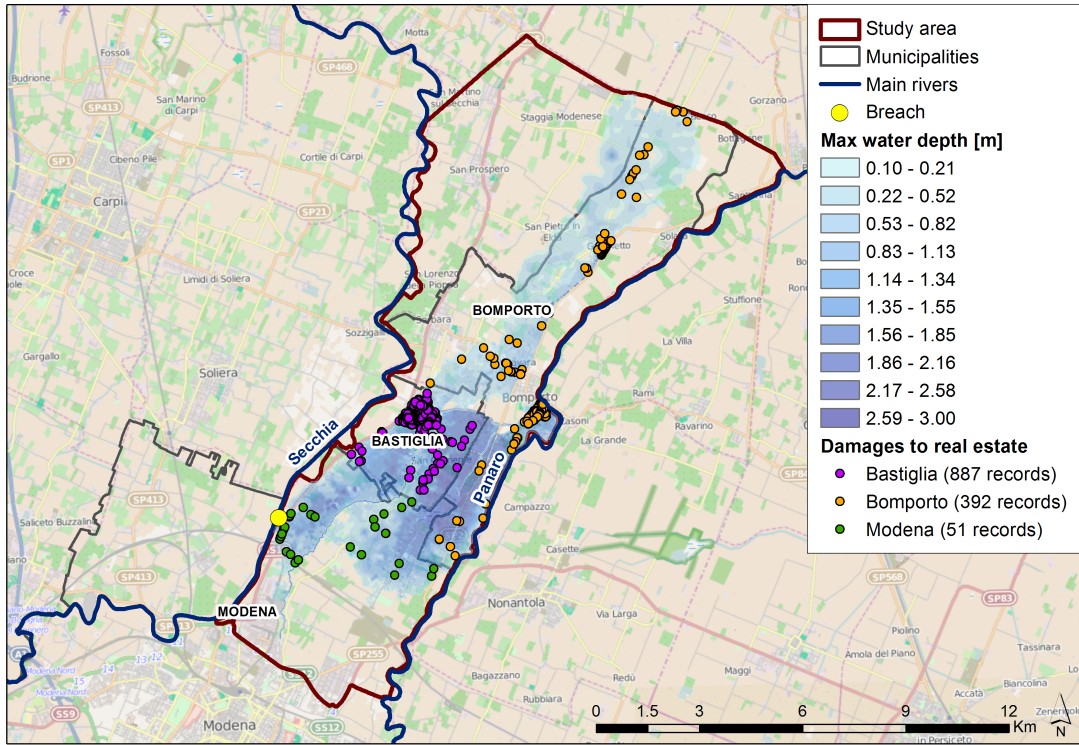

**Figure 2.** Maximum water depths simulated by the 2D model; geolocated buildings damages (colors reflect municipalities).




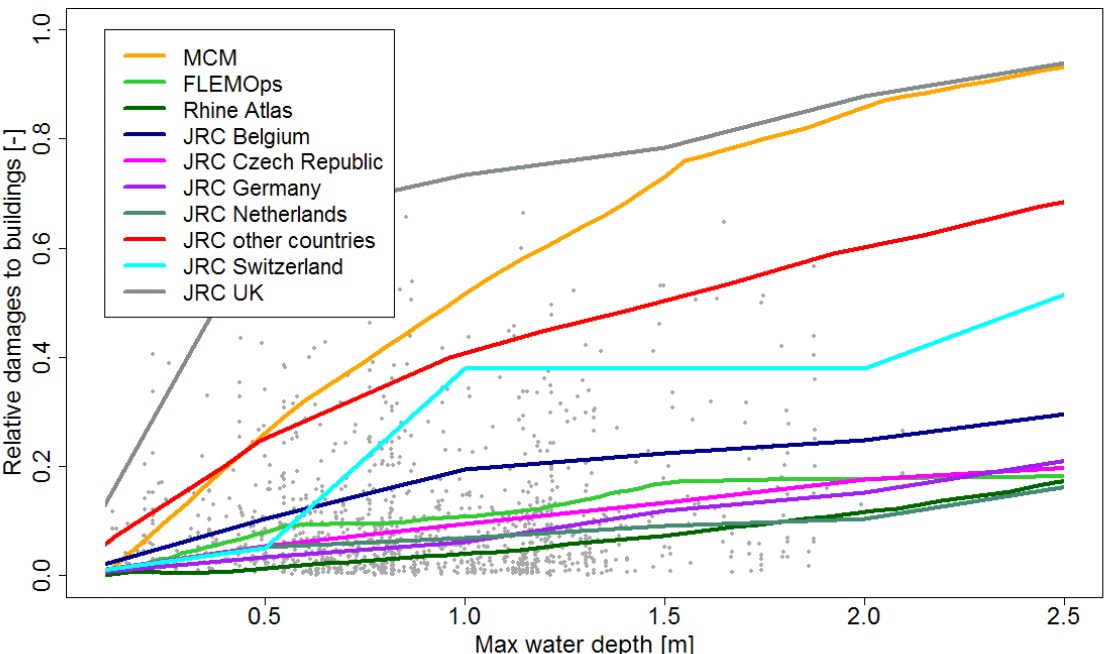

**Figure 3.** Literature stage-damage models and observed data: grey points in the background represent the observed relative loss (buildings only); literature models are limited to the maximum water depth reconstructed for the inundation event through the 2D hydrodynamic model (i.e. 2.5 m). Grey points in the background represent the observed relative loss (buildings only).




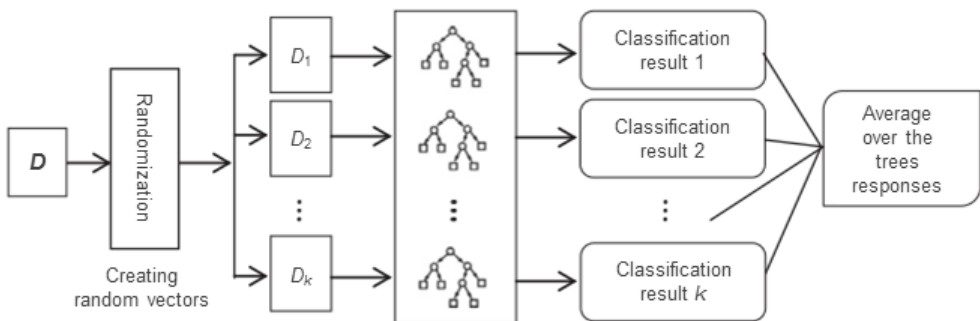

**Figure 4.** Random Forest method (Wang et al., 2015).





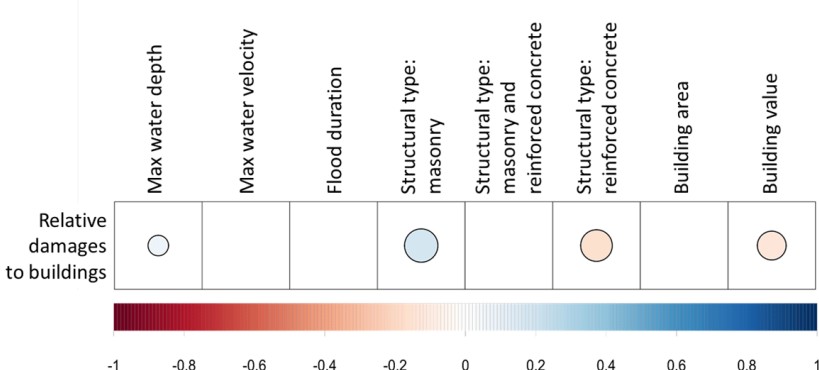

**Figure 5.** Spearman correlation between relative loss (buildings only) and predictive variables: maximum water depth; maximum water velocity; flood duration; structural type: masonry, masonry and reinforced concrete or reinforced concrete; building area; building value. Empty boxes indicate statistically non-significant correlation coefficients at a 5% significance level.





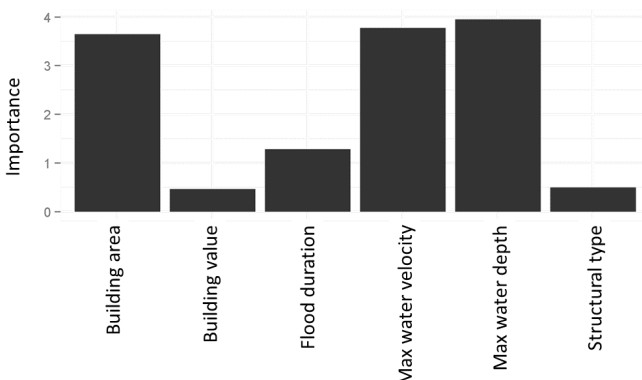

**Figure 6.** Importance of predictive variables considered in the MV model (building area; building value; flood duration; maximum water velocity; maximum water depth; structural type).





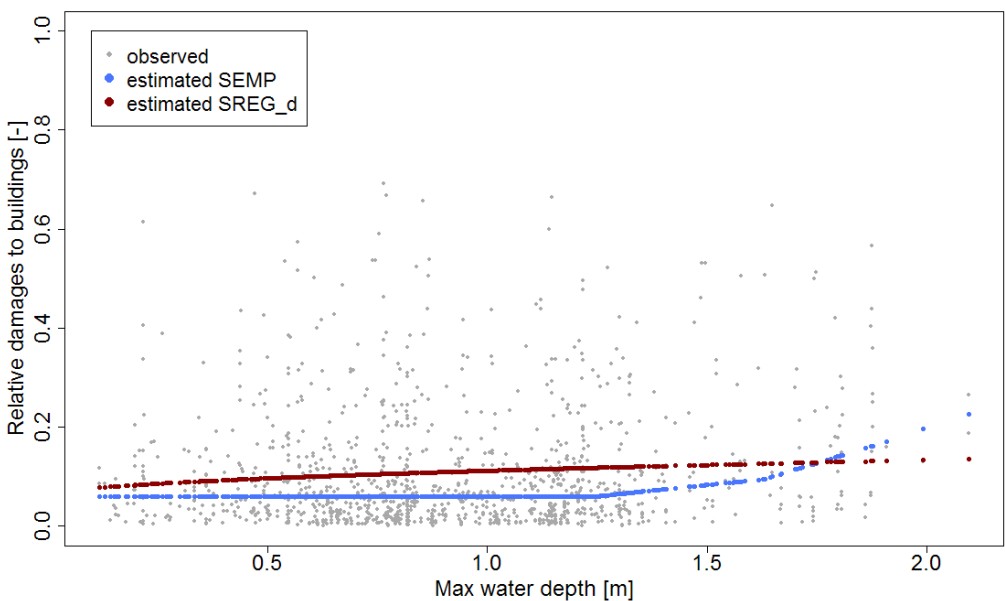

**Figure 7.** Relative damages to buildings estimated with the SEMP model (blue dots) and the $SREG_d$ model (dark red dots). Grey points in the background represent the observed relative loss (buildings only).





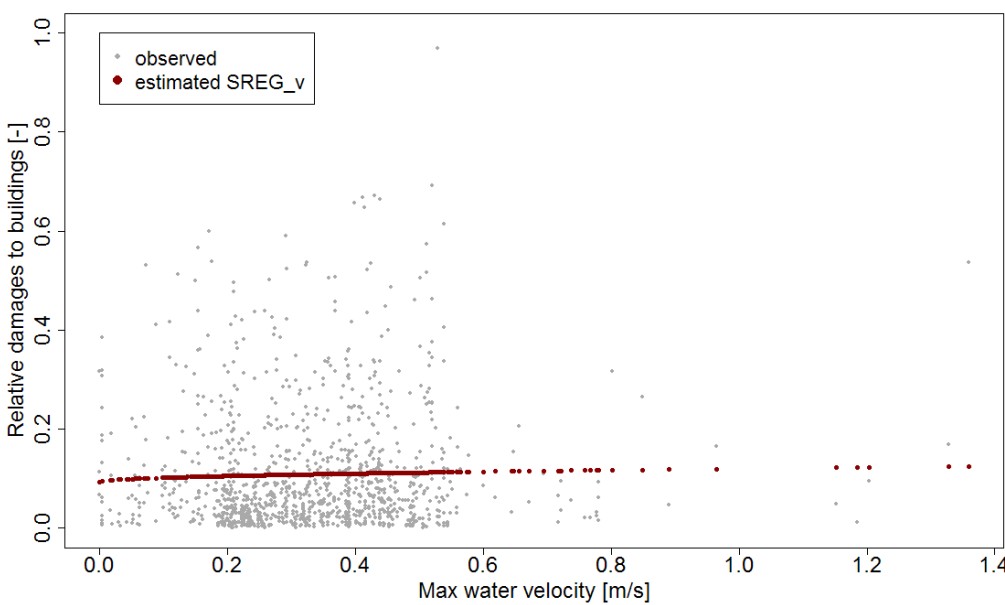

**Figure 8.** Relative damages to buildings estimated with the $SREG_v$ model (dark red dots). Grey points in the background represent the observed relative loss (buildings only).





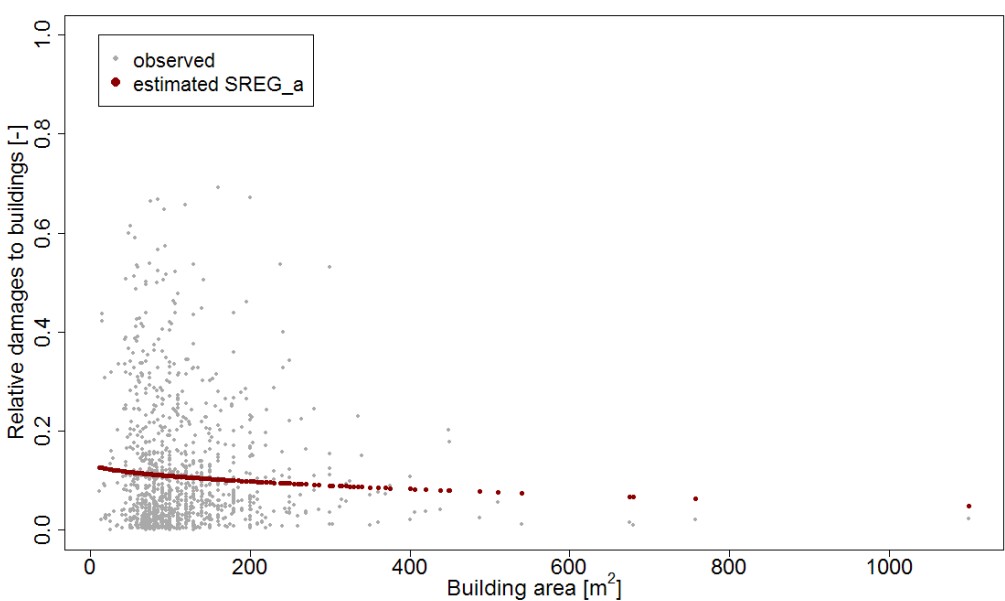

**Figure 9.** Relative damages to buildings estimated with the $SREG_a$ model (dark red dots). Grey points in the background represent the observed relative loss (buildings only).




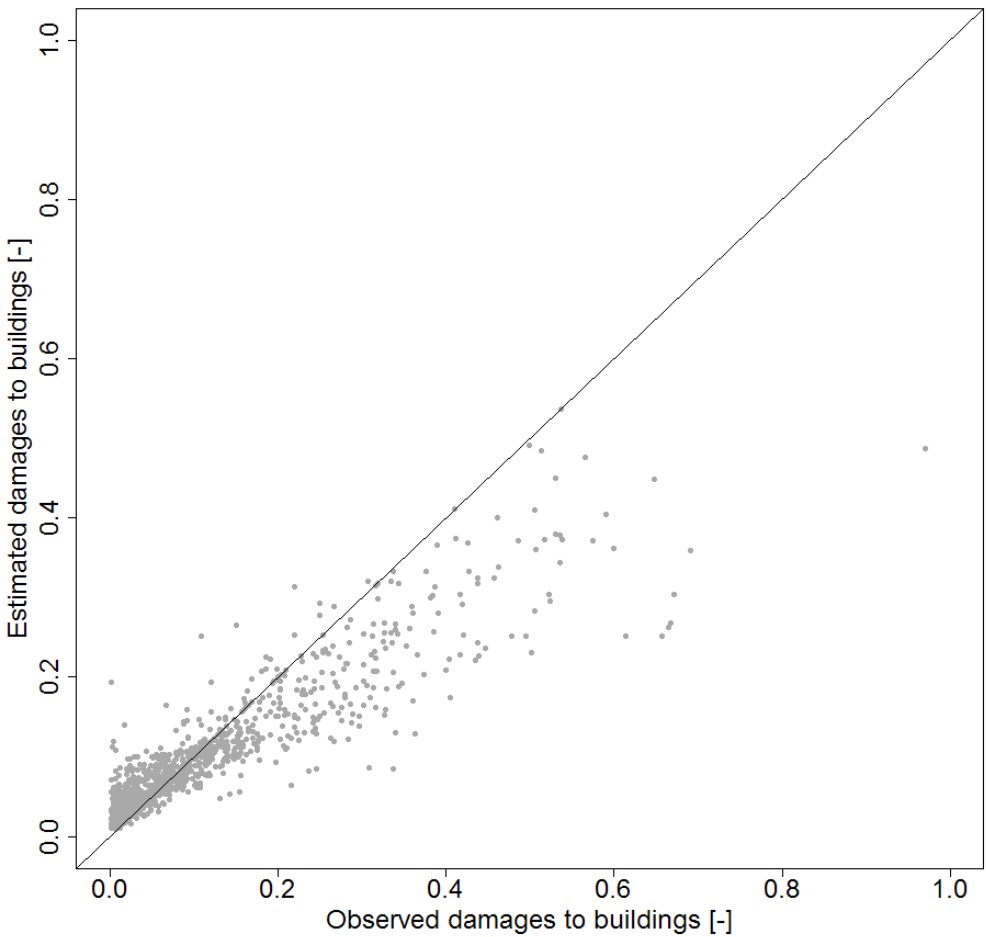

**Figure 10.** Relative damages to buildings estimated with the SMV model.




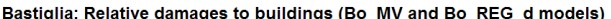

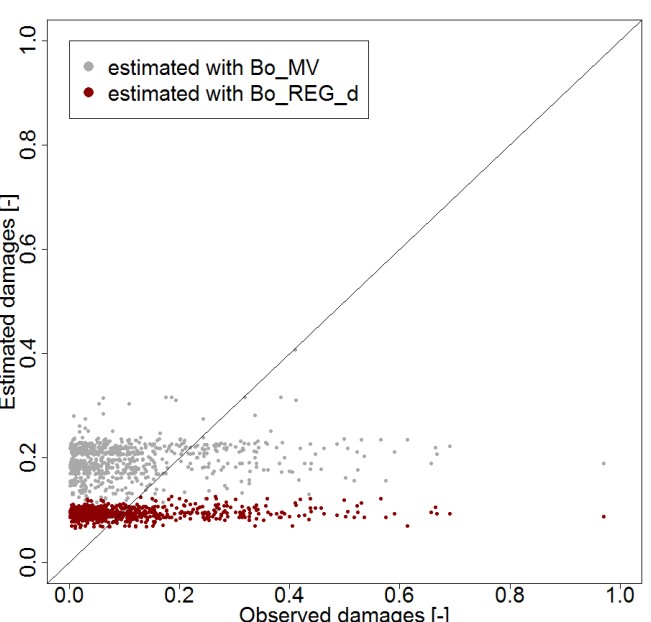

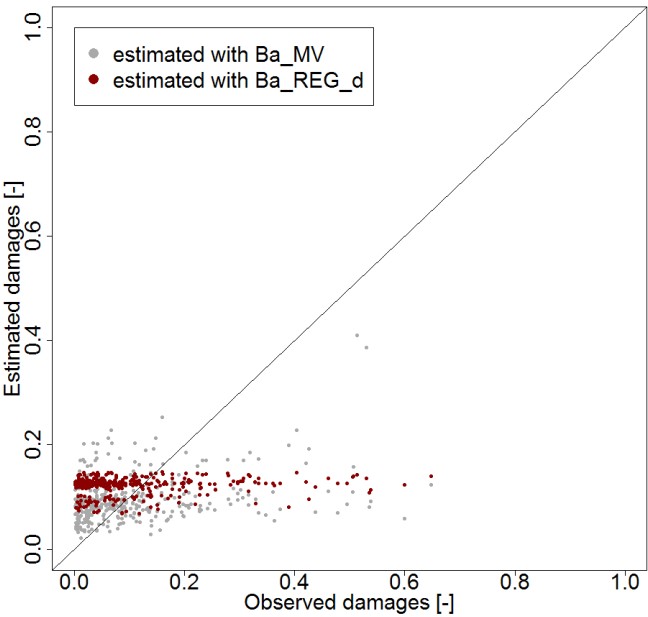

**Figure 11.** Top panel: Bastiglia relative damages to buildings estimated with $REG_d$ model (red dots) and the MV model (grey dots), both calibrated on Bomporto data set; Bottom panel: Bomporto relative damages to buildings estimated with $REG_d$ model (red dots) and the MV model (grey dots), both calibrated on Bastiglia data set.




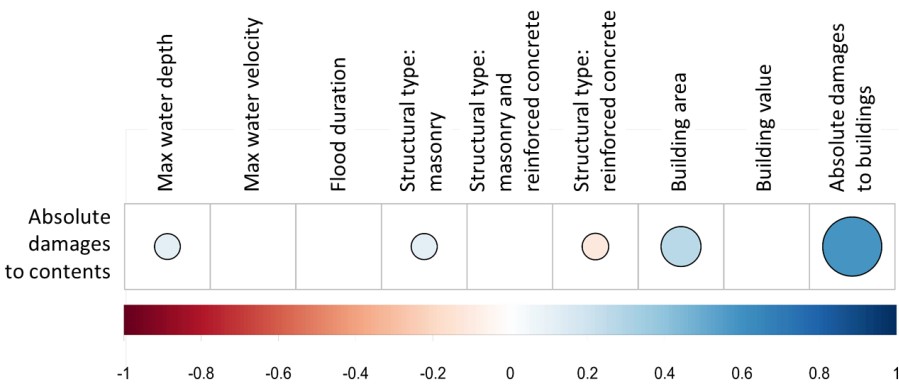

**Figure 12.** Spearman correlation between relative loss (contents only) and predictive variables: maximum water depth; maximum water velocity; flood duration; structural type: masonry, masonry and reinforced concrete or reinforced concrete; building area; building value. Empty boxes indicate statistically non-significant correlation coefficients at a 5% significance level.




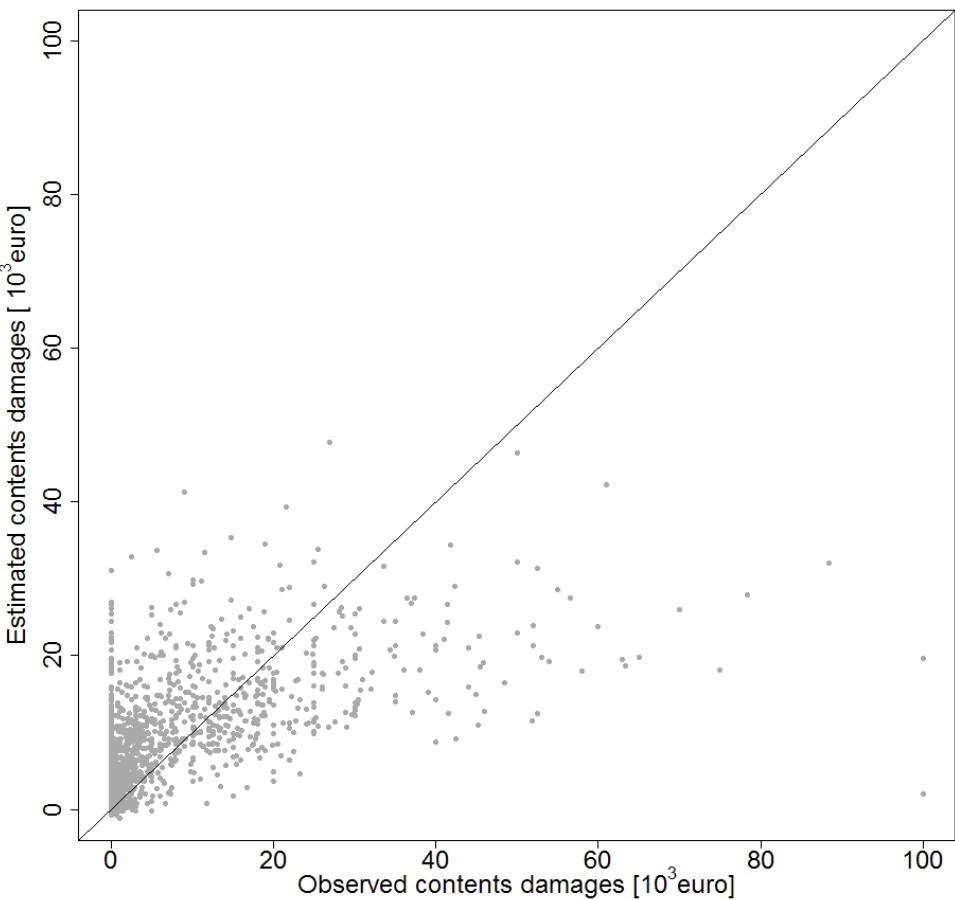

**Figure 13.** Empirical vs. predicted monetary losses to contents for the Secchia 2014 inundation event. Monetary loses are predicted as a function of monetary losses to building through Eq. 7.



## Tables

**Table 1.** Number of forms filled by private owners per municipality.

| Municipality | Affected private properties | Affected private properties (available address and at least damages to buildings) |
|:---:|:---:|:---:|
| Bastiglia | 1728 | 887 |
| Bomporto | 624 | 392 |
| Modena | 76 | 51 |
| **Total** | 2448 | 1330 |



**Table 2.** Refundable assets in accordance to Ordinance No. 2 of $5^{th}$ June 2014 and Law No. 93 of $26^{th}$ June 2014.

| Typology | Description | |
| --- | --- | --- |
| **Damages to buildings** | - Structural parts: | roofs, foundations, supporting structures, interior or exterior stairs, retaining walls for the stability of the building; |
| | - Non-structural parts: | walls or delimitation fence, interior flooring, plastering, interior and exterior painting, interior and exterior fixtures; |
| | - Installations: | electrical, heating, water, TV antenna, lifts, stair lifts for disabled or elderly people. |
| **Damages to contents** | - Furniture and household appliances: refrigerator, dishwasher, oven, sink, stove, washer, dryer, TV and personal computers. | |





**Table 3.** Considered variables and their sources and ranges, for buildings and contents damage analysis.

| Variable | Observed | Simulated | External sources | Range |
|---|:---:|:---:|:---:|:---:|
| Maximum water depth [m] | | ● | | 0.12 - 2.10 m |
| Maximum water velocity [m/s] | | ● | | 0 - 1.36 m/s |
| Flood duration [h] | | ● | | 2 - more than 30 h |
| Building area [m$^2$] | ● | | | 12 - 1100 m$^2$ |
| Building value [€/m$^2$] | | | ● | 902 - 1183 €/m$^2$ |
| Structural typology [-] | ● | | | masonry/reinforced concrete/combination of the two |
| Absolute damages to buildings [€] | ● | | | 40 - 158 659 € |
| Relative damages to buildings [-] | ● | | | 0 - 1 |
| Absolute damages to contents [€] | ● | | | 0 - 100 000 € |





**Table 4.** Performance of the uni- and multi-variable models developed on local data, in estimating relative damages to buildings. Models are ranked according to RMSE values, from the lowest to the largest. Correspondent results for literature models are reported in Table 5.

|  | BIAS [-] | MAE [-] | RMSE [-] | Differences between total estimated and total observed (€ 16.3 million) damages to buildings [%] |
|---|---|---|---|---|
| SMV | -0.012 | 0.034 | 0.062 | -9.1 |
| SREG$_d$ | 0.000 | 0.089 | 0.124 | 4.9 |
| SREG$_a$ | 0.000 | 0.089 | 0.124 | 1.2 |
| SREG$_v$ | 0.000 | 0.090 | 0.124 | 5.5 |
| SEMP | -0.043 | 0.080 | 0.130 | -34.0 |



**Table 5.** Performance of different literature uni-variable models in estimating relative damages to buildings. Models are ranked according to RMSE values, from the lowest to the largest. Correspondent results for uni- and multi-variable models developed on local data are reported in Table 4.

| | BIAS [-] | MAE [-] | RMSE [-] | Differences between total estimated and total observed (€ 16.3 million) damages to buildings [%] |
|---|---|---|---|---|
| FLEMOps | -0.003 | 0.089 | 0.125 | 1.8 |
| JRC Czech Republic | -0.022 | 0.085 | 0.127 | -15.2 |
| JRC Netherlands | -0.043 | 0.082 | 0.131 | -34.8 |
| JRC Germany | -0.046 | 0.082 | 0.133 | -37.2 |
| JRC Belgium | 0.056 | 0.119 | 0.142 | 53.7 |
| Rhine Atlas | -0.071 | 0.087 | 0.143 | -59.8 |
| JRC Switzerland | 0.149 | 0.196 | 0.232 | 137.2 |
| JRC other countries | 0.256 | 0.272 | 0.300 | 234.1 |
| MCM | 0.350 | 0.364 | 0.406 | 317.7 |
| JRC UK | 0.585 | 0.586 | 0.607 | 528.1 |





**Table 6.** Validation of the models: performance of the uni- and multi-variable models developed on two thirds of local data (randomly chosen) and validated on the remaining third of the records, in estimating relative damages to buildings. Models are ranked as in Table 4.

|  | BIAS [-] | MAE [-] | RMSE [-] |
|---|---|---|---|
| SMV | -0.022 | 0.084 | 0.127 |
| SREG$_d$ | -0.001 | 0.090 | 0.124 |
| SREG$_a$ | 0.000 | 0.090 | 0.124 |
| SREG$_v$ | 0.000 | 0.089 | 0.125 |
| SEMP | -0.042 | 0.081 | 0.131 |

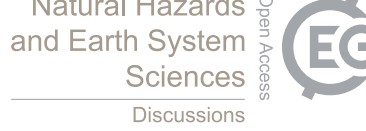

**Table 7.** Performance of different uni- and multi-variable models in estimating relative damages to buildings. In the upper tables, the models were calibrated on Bomporto's data set (392 records) and validated in Bastiglia, while in the bottom tables the models were calibrated on Bastiglia's data set (887 records) and used to estimated damages in Bomporto. Left tables report performance of the models in the calibration phase, while right tables show performance of the validation study.

| | *Calibration on Bomporto's data set (392 records)* | | | *Validation on Bastiglia's data set (887 records)* | | |
|---|---|---|---|---|---|---|
| | **BIAS [-]** | **MAE [-]** | **RMSE [-]** | **BIAS [-]** | **MAE [-]** | **RMSE [-]** |
| **Bo_MV** | 0.001 | 0.031 | 0.192 | 0.087 | 0.134 | 0.153 |
| **Bo_REG_d** | 0.000 | 0.085 | 0.118 | 0.007 | 0.089 | 0.127 |
| **Bo_REG_v** | 0.000 | 0.085 | 0.118 | 0.007 | 0.090 | 0.127 |
| **Bo_REG_a** | 0.000 | 0.085 | 0.118 | 0.007 | 0.089 | 0.127 |

| | *Calibration on Bastiglia's data set (887 records)* | | | *Validation on Bomporto's data set (392 records)* | | |
|---|---|---|---|---|---|---|
| | **BIAS [-]** | **MAE [-]** | **RMSE [-]** | **BIAS [-]** | **MAE [-]** | **RMSE [-]** |
| **Ba_MV** | -0.012 | 0.040 | 0.071 | -0.004 | 0.080 | 0.113 |
| **Ba_REG_d** | 0.000 | 0.091 | 0.126 | 0.007 | 0.087 | 0.118 |
| **Ba_REG_v** | 0.000 | 0.091 | 0.126 | 0.007 | 0.088 | 0.118 |
| **Ba_REG_a** | 0.000 | 0.091 | 0.126 | 0.007 | 0.088 | 0.118 |





**Table 8.** Performance of different uni- and multi-variable models in estimating damages to contents via Eq. 7. Models are ranked according to RMSE values, from the lowest to the largest.

| | BIAS [€] | MAE [€] | RMSE [€] | Differences between total estimated and total observed (€ 11 million) damages to contents [%] |
|---|---|---|---|---|
| *Obs. buildings loss* | *0* | *6 790* | *10 742* | *7.0* |
| JRC Netherlands | 299 | 8 993 | 12 702 | -0.9 |
| SEMP | -349 | 8 769 | 12 703 | -0.1 |
| JRC Germany | -491 | 8 722 | 12 708 | -5.5 |
| JRC Czech Republic | 2 051 | 9 684 | 12 863 | 15.5 |
| Rhine Atlas | -2 528 | 8 174 | 12 948 | -32.7 |
| $SREG_v$ | 2 903 | 1 0066 | 13 026 | 34.5 |
| FLEMOps | 3 121 | 10 167 | 13 076 | 30.0 |
| $SREG_a$ | 3 362 | 10 283 | 13 136 | 32.7 |
| $SREG_d$ | 3 445 | 10 324 | 13 157 | 34.5 |
| JRC Belgium | 7 671 | 12 705 | 14 836 | 65.5 |
| SMV | 8 520 | 13 246 | 15 292 | 14.5 |
| JRC Switzerland | 14 481 | 17 634 | 19 260 | 11.4 |
| JRC other countries | 16 260 | 19 051 | 20 631 | 103.6 |
| MCM | 19 365 | 21 659 | 23 157 | 184.5 |
| JRC UK | 25 996 | 27 527 | 28 931 | 260.9 |