# Peer review of "Development and assessment of uni- and multi-variable flood loss models for Emilia-Romagna (Italy)"

_Natural Hazards and Earth System Sciences, 2017_

## Referee Comment (RC1) · Anonymous Referee #1 · 4 Nov 2017

Review of the manuscript NHESS-2017-342 The paper describes the development of flood loss models on the basis of a remarkable dataset of observed flood losses. This dataset was used to develop different kinds of loss models and to validate these models, as well as other models available in the literature. In general, the paper presents an interesting study. The novelty lays in the approach for developing a new approach for multi-variable flood loss models. However, while reading the paper, some questions arose. With some explanations added, the paper will be of interest for the flood loss modelling community. The main and principle question that arises is, if the random forest approach is sensitive to heteroscedasticity in the data. As figure 10 shows, the deviations from the observed data vary with magnitude. It is highly recommended to

test the data for heteroscedasticity and to tackle with this issue in the development of the models if necessary. It would be of interest how the residuals are distributed. Furthermore, as in the introduction is stated, the slope of the floodplain is very regular. Thus, flow depths vary only in case of backwater effects of hydraulic obstacles. Hence, flow velocities in this relatively homogeneous case study may not be considered as independent variables (dependent on flow depth). I don't know how the flood model used for the analysis computes velocity and flow depth. Anyway, they are interlinked through the model used. However, this is a hypothesis and the contrary should be demonstrated. While looking at Fig. 11, a question arises if both cases Bastiglia and Bomporto do have relatively homogeneous flow depths inside of their samples but differs remarkably between both. This may lead to an overrepresentation of a certain flow depth interval and hampers the transferability of a model calibrated on one case study to the other case study. Figure 1 strengthens this observation, although the flow depths are not visible below the clustered points. I recommend showing a box plot of the flow depths at the single buildings for both case studies. The authors are asked to assess the reliability of the flood loss estimations (in monetary terms) by the home owners immediate after the flood event. I suspect that all home owners have the competency for estimating the damages to their buildings as professionals have (insurance experts and craftsmen commissioned to restore the building). The authors should describe how these estimations were "verified for authenticity" by the administration. If this verification was made following a reliable approach, the refunded value should be used for the analyses and not the estimations. Another weak point is the use of the market value for the estimation of the building's values. It is not described, if this value comprises the cost of the land too. Furthermore, it is not documented if this value is given for the area of the building footprint or for the living space that should be multiplied by the number of floors. The comparison between different flood loss models should consider the used base value for assets. It would be of interest which approach the authors followed for the geolocation of the loss data. p. 8, ln. 19: is the size of 1 to 200 m for element length or area of the element? p-11, ln. 26 chapter 4.2.1. It is not

defined what "best performance" means here. Results section. The model structure of the multi-variate model, i.e. the outcomes of the random forest analysis, should be described. Which parameter with which weights have been identified and structure the prediction model. In its present form, the reproducibility it is not given. One solution could be to adapt Fig. 4 and insert the resulting model structure. p. 14, ln. 28-29. In addition to the comparison of the predicted losses with observed ones, it would be of interest splitting the dataset stochastically. Together with the comparison between both calibration datasets with the opposite case study data, the conclusion of the transferability could be grounded more reliably. A sensitivity test of the SMV model should be done. p.17, ln.1-5. There is a conflict between text and figure 11. In the text, the grey dots are described as observations. In the figure, no blue dots are visible as mentioned in the text. p.17, ln.10. "in the sake of brevity". This can be shown in the appendix p.18, ln. 16. What is "Sec. 8"? Fig. 1: The authors are asked to explain why they mapped only flow depths >10 cm. Are the analyses based on the full range of flow depths or are flow depths >10 cm generally omitted throughout the study?

---

## Referee Comment (RC2) · Anonymous Referee #2 · 6 Nov 2017

Journal: NHESS
Title: **Development and assessment of uni- and multi-variable flood loss models for Emilia-Romagna (Italy)**
Author(s): Carisi et al.
MS No.: NHESS-2017-342
MS Type: Research Article
**Iteration: First review**

The paper addresses flood loss estimation in Northern Italy, trying to highlight possibilities and limitations. By using flood damages recorded after the flood of the Secchia river in 2014, the authors (i) derive uni- and multi-variable damage models for the study area and compare them with models from the literature (ii) evaluate the transferability of such models to similar contexts and finally (iii) explore the relationship between damage to buildings and damage to contents for the available dataset.

The paper is in the scope of the journal and of interest for the research community working on flood risk; although "local" in the analyses, its results can be generalised to other contexts as well.

The paper is well organised, data are properly described, as well as methods, although some minor integrations/specifications are required with respect to the latter. Likewise, there are some minor imprecisions to be corrected in the whole text. The discussion of results can be improved with respect to some aspects (see below).

In general, the paper is a little bit long. Some suggestions are provided in the following on parts that can be neglected or shortened; nonetheless, the paper can take advantage of an English review aimed at simplifying articulated and (repetitive) sentence.

**Major criticisms**

Section 1
- The Introduction is too long. I would shorten the first paragraphs on the importance of flood losses and omit the discussion on aleatory and epistemic uncertainty (the following part on specific uncertainties related to damage models is more interesting for the paper).
- Section 1.1 should be re-organised by first declaring the objectives of the research and then the tools/methods. The present form is totally clear only after reading the whole paper.

Section 3.1
- The discussion on the difference between declared and refunded damage can be shortened in my opinion, by neglecting details.
- I agree on the use of declared data (instead of refunded damages) but it is not clear whether implemented damage data above 15.000 euros were verified or not. If this is the case, data below 15.000 euros are less reliable and authors should take this aspect into account in the analysis.
- I do not agree with the use of OMI data for the assessment of buildings value that, as stated by the authors in the Conclusions, "are more an expression of the overall economic well-being of a specific area" rather than of the real value of the buildings. (Re)construction costs are more suitable to the objective in my opinion.

Section 4.1
- The description of the damage models can be shortened by referring to available literature and leaving only the significant information for the paper (i.e. how models have been implemented).
- Authors implement models developed to be applied at the micro-scale (e.g. MCM, Flemo-PS) and models developed to be applied at the meso-scale (e.g. Rhine Atlas, JRCs). I guess whether damage estimation (i.e. models performance) is influenced by the different levels of knowledge/detail of input variables required by the models vs. available data. Did authors explore this aspect?

*Section 4.1.1*

- How authors converted the absolute curves of MCM in relative curves? MCM curves were developed in 2005 while the flood occurred in 2014; Did authors apply a discount rate to estimated damage?
- Why authors chose to convert absolute curves by mean of the average economic building value in the study area rather than by using different values for the different OMI zones? I would adopt this second option as MCM is a "micro" scale damage model.

Section 4.2
- Which is the formulation of SEMP?

Section 5.1
- From figures 7, 8, 9, it seems that uni-variable local models always estimate a relative damage around 0.1 (independently of the value of the dependent variable). Did authors notice that? How it can be justified?
- How authors justify the bad performance of SVM in estimating the total absolute damage?
- With regard to existing models, I expect that models with the best performance underestimate the total damage (as citizens tend to overestimate damage during declaration). In fact, four of the six best models underestimate. Can authors comment on that?

Section 5.2
- This section could be rewritten and improved to better explain the significance of results. Finding correspondence between authors' considerations and figures/tables is not straightforward at present.
- There is no correspondence between Figure 11 and its description in the text. Check also models acronym. Correspondence between test and figures is often lacking.

Section 5.3
- The link between the performance in estimating damage to buildings and damage to contents is not so evident to me. Why SMV that is the one with the best performance in estimating damage to buildings is quite bad in estimating damage to contents?

Conclusions
- The transferability of local models stated in the last part of the section should be better discussed previously in the paper. Two/three sentences highlighting this point can make conclusions more robust

NB
Pay attention to be consistent in terminology. Authors use damage to "contents" and "content" interchangeably. I guess they are typos. The same can be state for model acronyms (e.g. SMV sometimes becomes MV).

**Specific minor comments (which can increase the readability and clarity of the paper)**

Section 1

Pg. 2 line 17 "flood risk is the combination of hazard (i.e. the probability of a flood event with a certain intensity to occur in a specific area and in a specific time period) and consequences, providing for instance information on the vulnerability, i.e. the type and number of elements affected by a given flood event, and how well they are able to resist" → from this statement, I understand that consequences and vulnerability are the same "concept", please rephrase

Pg. 2 line 24 "Uncertainty exists in all flood risk components" → do authors mean "in the estimation of" all risk components?

Pg. 2 line 35 "Nevertheless, several authors indicate that damage models still provide an important sources of uncertainty in flood damage estimates" → do authors mean in flood "risk" estimates?

Pg. 3 line 12 "These models were shown to outperform uni-variable loss models, under the condition that sufficiently large and detailed damage data-sets are provided" → for what? Please specify

Pg. 3 line 16 "A further aspect that contributes to the uncertainty is the lack of sufficient, comparable and reliable high quality flood loss data" → for what? Please specify

Pg. 3 line 17 "In the absence of empirical damage data, damage models are either selected from the literature or subjectively and schematically derived by experts using a synthetic approach" → I would move this sentence after line 32, i.e. after the discussion on the importance of data

Pg. 3 line 35 "see e.g. Molinari et al., 2014b, on the transferability of the model developed on the basis of specific flood event data by Luino et al. (2006) and Freni et al. (2010)" → the content of the paper authors refer to is not clear, please rephrase

Pg. 4 line 8 → add the year of the event

Pg. 4 line 11 "The raw data collected by local authorities has been homogenized, geocoded and integrated with other useful information" → while geocoding and integration are fully described in the paper, homogenization is not discussed. Please give more information on this aspect.

Pg. 4 line 18 "Second, we calibrate empirical uni- and multi-variable models to subsections of the study area and validate them using the data observed in different subsections (split-sample validation)" → not clear at this point of the paper, please rephrase

Section 2

Pg. 5 line 7 "the aspect of the area is oriented" → I am not sure "aspect" is the right term here

Pg. 5 line 10-13 → and what about the southern boundary? How was it chosen?

Pg. 5 line 21 "Thanks to several eyewitness accounts, video footage and studies conducted by the scientific committee" → Which scientific committee? Please explain

Section 3

Pg. 5 line 31 "citizens and property owners were asked to fill forms about public properties damages (form A)" → according to my knowledge form A is compiled by public authorities

Pg. 6 line 5 "The database regarded private properties affected by different kinds of potential damages" → why potential?

Pg. 7 line 22 "Focusing on residential buildings, we defined the building's economic values [e/m2] as the average of the values provided for each property in the same OMI zone" → Did authors consider 2014 values?

Section 4

Pg. 9 line 10 "These models associate relative (or absolute) losses with different input variables. The most frequently used models in Europe are uni-variable damage models, i.e. they estimate the amount of relative damages as a function of a single input variable, most commonly water depth, (Merz et al., 2010; Messner et al., 2007; Jongman et al., 2012), differentiated by building use, type, etc. (Gerl et al., 2016). → this aspect has been already discussed in the Introduction and can be omitted here

Pg. 9 line 20 "The damage curve implemented in the Multi-Coloured Manual" → I guess damage "curves"

Pg. 13 line 14 "The variables being randomly permuted presenting a low accuracy are the most important ones in the damage prediction, as their influence in the prediction process is very high" → not clear, please rephrase

Section 5

Pg. 14 line 5 "dummy variable encoding" → please check

Pg. 14 line 8 "The only variables that resulted significantly correlated with the relative loss to buildings were the maximum water depth, building value and structural typology. However, correlations coefficients between these variables and relative damages are low, precisely lower than +/-0.18" → where I can see this value?

Pg. 15 line 13 "Although this values are satisfying in terms of errors, the performance of this models are lower than the ones of the models developed on Secchia's data set (except SEMP model)" → Why authors can state this? A deeper discussion at this point can help the reader also in the following analyses

Pg. 15 line 28 "The reason behind this fact must be attributed to the morphologic and socio-economic context where this models have been drown, that differs considerably from the Secchia ones, in addition to the different criteria adopted to develop them" → which criteria authors refer to? Please comment

Pg. 17 line 31 "damages to contents turned out to be significantly correlated with the building footprint (Spearman correlation coefficient equal to 0.27) instead of the building value. A noteworthy feature of Figure 12 is the very strong and statistically significant positive correlation between damages to buildings and to their content (Spearman correlation coefficient equal to 0.59)" → Where I can see values of the correlation coefficients?

Pg. 18 line 21 "The performance of all considered models, with the exception of the last four in Table 8, show a difference between observed and predicted overall monetary losses to contents that does not exceed e 4 million (except for JRC Belgium that presents a difference value of e 7.2 million)" → These values refer to the bias, i.e. the mean difference between observed and estimated values, not to the difference between observed and predicted overall monetary losses

Section 6

Pg. 19 line 18 "the building values provided by the Italian Revenue Agency (Agenzia delle Entrate - AE) represent the buildings market values at a given time of given building typologies that is more an expression of the overall economic well-being of a specific area rather than the depreciated economic buildings values in case of a flood event " → I agree that OMI values are not representative of buildings value but I do not agree on the need of considering depreciated values. Please comment

Pg.19 line 22 "the multi-variable (MV) forest provide the important advantage to avoid the need to find a parametric function that works with all the data" → not clear, please comment

Pg. 19 line 24 "These can be very useful information for authorities and stakeholders to define preventive measures and/or mitigation strategies " → some examples should increase the robustness of the statement

Figures

Figure 3 - caption "Grey points in the background represent the observed relative loss (buildings only)" → the sentence is repeated

Figure 4 is not recalled in the paper

Figure 6 - caption → please, correct MV with SMV

Figure 12 – caption→ "absolute damage to buildings" is missing from the list of predictive variables

Figure 13 → try to convert in logarithmic axes, the readability can improve

Table 2 → what are "interior and exterior fixtures"

Bibliography

I did not check the bibliography at this stage of the review. I reserve to do this in a second time.

---

## Author Response (AR1)

DEVELOPMENT AND ASSESSMENT OF UNI- AND MULTI-VARIABLE FLOOD LOSS MODELS FOR EMILIA-ROMAGNA (ITALY)

by Francesca Carisi, Kai Schröter, Alessio Domeneghetti, Heidi Kreibich, Attilio Castellarin

**REPLY TO EDITORS' AND REVIEWERS' COMMENTS**

We would like to sincerely thank the Editor for her review and the possibility to improve the quality of the manuscript, also granting additional time to perform further analyses. We also sincerely acknowledge the very useful and insightful comments and suggestions raised by both Reviewers. Our revised manuscript addresses all major and minor comments raised during the reviewing process, following the Editor's indication in case of conflicting comments from the Reviewers.

The rest of the document uses the following notation:

- Black: original comments from Reviewers and Editor
- Blue: our original replies during the discussion phase
- Red: actual revisions implemented in the revised manuscript, together with an explicit indication to the revised parts in the manuscript (i.e. lines and pages of the revised manuscript), when applicable.

**EDITOR DECISION:**

Reconsider after major revisions (further review by editor and referees)

(04 Feb 2018) by Margreth Keiler
Comments to the Author:

Dear Francesca Carisi and co-authors,

Thank you very much again for your submission "Development and assessment of uni- and multi-variable flood loss models for Emilia-Romagna (Italy)". Both referees acknowledged that you have taken up a timely and interesting topic of addressing flood loss estimation. However, both referees and I agree that reading the current manuscript leads to lot of open questions which indicate that the manuscripts needs improvements before we can consider your manuscript for publication. Therefore, a major revision of the manuscript is necessary.

Both reviewers provided a detailed reports highlighting all the points you should address in the revisions. According to your response, I am very positive about the new version of the manuscript and that you will take up their remarks. I see that you have the challenge about the contradicting remarks of the reviewers to shorten the manuscript (indeed it is very long) and to be asked for more details. I suggest to follow reviewer 2 regarding section one and in general, but provide the more details in section 3.

Please also note that this decision does not necessarily imply acceptance of the manuscript in the journal NHESS, and it still will depend on your reply (and subsequent edits to your manuscript) to referees comments, as well as on the reviewer comments of the revised version.

I look forward to receiving the revised version of your manuscript.

Regards, Margreth Keiler

NHESS Editor

Associate Professor of Geomorphology, Natural Hazards and Risk Research, University of Bern

Many thanks for your additional assessment of out manuscript. We followed your suggestions and addressed all comments raised by reviewers, as described below.

**ANONYMOUS REFEREE #1**

The paper describes the development of flood loss models on the basis of a remarkable dataset of observed flood losses. This dataset was used to develop different kinds of loss models and to validate these models, as well as other models available in the literature. In general, the paper presents an interesting study. The novelty lays in the approach for developing a new approach for multi-variable flood loss models. However, while reading the paper, some questions arose. With some explanations added, the paper will be of interest for the flood loss modelling community.

We would like to sincerely thank the Anonymous Referee #1 for his positive review and input, which helps us significantly in improving the presentation of our study.

The main and principle question that arises is, if the random forest approach is sensitive to heteroscedasticity in the data. As figure 10 shows, the deviations from the observed data vary with magnitude. It is highly recommended to test the data for heteroscedasticity and to tackle with this issue in the development of the models if necessary. It would be of interest how the residuals are distributed.

We thank the Reviewer for pointing this aspect out, which we missed to properly address in the original version of the manuscript. We will deepen the analyses about heteroscedasticity, performing the tests suggested, in order to improve the quality and the robustness of our multi-variable model, and we will examine which measure can be taken to correct it where appropriate.

After some additional review of the literature and basic material on random forest approach, we can state that heteroscedastic errors are not of concern for it. Random forest algorithm doesn't include any ordinary least square based pruning, so it is not affected by this problem. In fact, another important advantage of this algorithm is that no assumptions about independence, distribution or residual characteristics are needed. We specified it in the explanation of the multi-variable model (see p. 13, l. 3-4).

Furthermore, as in the introduction is stated, the slope of the floodplain is very regular. Thus, flow depths vary only in case of backwater effects of hydraulic obstacles. Hence, flow velocities in this relatively homogeneous case study may not be considered as independent variables (dependent on flow depth). I don't know how the flood model used for the analysis computes velocity and flow depth. Anyway, they are interlinked through the model used. However, this is a hypothesis and the contrary should be demonstrated.

For tree based models no assumption about independence of variables is needed. Anyway, as we are looking at the maxima both in case of water depth and velocity, they commonly refer to different time steps. Thus, we think it is not a problem if the descriptors show some degree of correlation. We will add a short explaining comment to the text.

Done (see p. 8, l. 21-22 and p.13, l. 3-4).

While looking at Fig. 11, a question arises if both cases Bastiglia and Bomporto do have relatively homogeneous flow depths inside of their samples but differs remarkably between both. This may lead to an overrepresentation of a certain flow depth interval and hampers the transferability of a model calibrated on one case study to the other case study. Figure 1 strengthens this observation, although the flow depths are not visible below the clustered points. I recommend showing a box plot of the flow depths at the single buildings for both case studies.

We agree with the Reviewer. In fact, water depths in Bastiglia are lower than in Bomporto, although the distributions of the observed damages are quite similar (as you can see in the box plots below). We agree that this is worth specifying it in the discussion of the results on model transferability.

[Figure]

[Figure]

Done (see p. 17, l. 6-9).

The authors are asked to assess the reliability of the flood loss estimations (in monetary terms) by the home owners immediate after the flood event. I suspect that all home owners have the competency for estimating the damages to their buildings as professionals have (insurance experts and craftsmen commissioned to restore the building). The authors should describe how these estimations were "verified for authenticity" by the administration. If this verification was made following a reliable approach, the refunded value should be used for the analyses and not the estimations.

We will improve the clarity of the section where we explain our choice to consider as observed losses the damages as claimed by citizens in Form B, instead of the refunds. Due to the specific and strict compensation criteria (i.e. not all damage is compensated) the refunded amounts differ from the "actual" damage.

Done (see p. 6, l. 3-23).

Another weak point is the use of the market value for the estimation of the building's values. It is not described, if this value comprises the cost of the land too.

The study assesses flood damages to buildings, in particular to their structural part and their contents. The use of the economic value of the structural part of the building, that doesn't take into account the land cost, is therefore congruent with the goal of the analysis. This will be clarified in the text.

Done (see p. 6, l. 34-35).

Furthermore, it is not documented if this value is given for the area of the building footprint or for the living space that should be multiplied by the number of floors.

Only the first floor of each building has been considered, being the maximum water depth lower than 2.5 m. This will be better explained in the revised manuscript.

Done (see p. 7, l. 1-2).

The comparison between different flood loss models should consider the used base value for assets.

We do not completely agree with this suggestion, because the models use damages, that are relativized based on each different context, therefore they are comparable to each other.

We specified it in the revised manuscript (see p. 8, l. 29-31)

It would be of interest which approach the authors followed for the geolocation of the loss data.

Thanks, we will improve and detail its description in the revised manuscript.

p. 8, ln. 19: is the size of 1 to 200 m for element length or area of the element?

The cited size refers to the length of the triangular elements of the computational mesh, we will clarify it in the revised text.

p-11, ln. 26 chapter 4.2.1. It is not defined what "best performance" means here.

It refers in particular to the Root Mean Square Error, it will be clarified in the revised manuscript.

Results section. The model structure of the multi-variate model, i.e. the outcomes of the random forest analysis, should be described. Which parameter with which weights have been identified and structure the prediction model. In its present form, the reproducibility it is not given. One solution could be to adapt Fig. 4 and insert the resulting model structure.

Thank you for your suggestion. Unfortunately, the structure of a Random Forest (RF) is difficult to describe. A RF consists of 500 bootstrap replica of each record of the dataset with one tree grown for each replica. RF are black-boxes and it is not possible to report each tree including details about all splits. We will show examples of built trees (perhaps in an Appendix), i.e. adapting Fig. 4. Additionally, we will use the appendix to detail the algorithm.

We added an example of a built tree for the Secchia case study in the Appendix (see Fig. A1) and we provided the reference for the detailed algorithm, in order to make the procedure even clearer (see p. 11, l. 31).

p. 14, ln. 28-29. In addition to the comparison of the predicted losses with observed ones, it would be of interest splitting the dataset stochastically. Together with the comparison between both calibration datasets with the opposite case study data, the conclusion of the transferability could be grounded more reliably. A sensitivity test of the SMV model should be done.

The Random Forest algorithm includes a stochastic splitting of the data by using bootstrap replica of the dataset to learn the individual trees of the forest. The predictions of these trees are aggregated to a common prediction. A sensitivity test of variables included in the SMV model is done in terms of the analysis of variable importance (cf. Figure 6), with higher importance values for more sensitive variables.

p.17, ln.1-5. There is a conflict between text and figure 11. In the text, the grey dots are described as observations. In the figure, no blue dots are visible as mentioned in the text.

Thanks, this error will be corrected (grey dots refer to the estimation of relative loss using the MV models).

Done (see p. 16, l. 16-17).

p.17, ln.10. "in the sake of brevity". This can be shown in the appendix

Good suggestion, we will keep it in consideration and add it in the appendix of the revised manuscript.

Done (see Fig. B1 and B2) in the Appendix.

p.18, ln. 16. What is "Sec. 8"?

It will be corrected (Sec. 5.1).

Done (see p. 18, l. 8).

Fig. 1: The authors are asked to explain why they mapped only flow depths >10 cm. Are the analyses based on the full range of flow depths or are flow depths >10 cm generally omitted throughout the study?

In order to take the uncertainties of hydrodynamic modelling into account, we regarded as flooded only those areas with simulated water depths above 10 cm. This will be better explained in the revised manuscript, also providing the reader with references.

Done (see p. 8, l. 16-17).

**ANONYMOUS REFEREE #2**

The paper addresses flood loss estimation in Northern Italy, trying to highlight possibilities and limitations. By using flood damages recorded after the flood of the Secchia river in 2014, the authors (i) derive uni- and multi-variable damage models for the study area and compare them with models from the literature (ii) evaluate the transferability of such models to similar contexts and finally (iii) explore the relationship between damage to buildings and damage to contents for the available dataset.

The paper is in the scope of the journal and of interest for the research community working on flood risk; although "local" in the analyses, its results can be generalised to other contexts as well.

The paper is well organised, data are properly described, as well as methods, although some minor integrations/specifications are required with respect to the latter. Likewise, there are some minor imprecisions to be corrected in the whole text. The discussion of results can be improved with respect to some aspects (see below).

In general, the paper is a little bit long. Some suggestions are provided in the following on parts that can be neglected or shortened; nonetheless, the paper can take advantage of an English review aimed at simplifying articulated and (repetitive) sentence.

The positive review and all specific remarks of Anonymous Referee #2, particularly the suggestions for a modification of the revised manuscript structure, are gratefully acknowledged and we will definitely take them into account, in order to reach a better presentation of our analysis.

Major criticisms

Section 1

- The Introduction is too long. I would shorten the first paragraphs on the importance of flood losses and omit the discussion on aleatory and epistemic uncertainty (the following part on specific uncertainties related to damage models is more interesting for the paper).

We will review and shorten the introduction, according to these suggestions.

Done.

- Section 1.1 should be re-organised by first declaring the objectives of the research and then the tools/methods. The present form is totally clear only after reading the whole paper.

Thank you for the advice, we will definitely follow it in the revised manuscript.

We incorporated this subsection in the introduction, re-organizing its structure as suggested (see p. 3, l. 23 – p. 4, l. 11).

Section 3.1

- The discussion on the difference between declared and refunded damage can be shortened in my opinion, by neglecting details.

Ok, we will take this comment into consideration, although a compromise is needed with the request of Anonymous Referee #1, who asks for a more detailed explanation of this part.

Done. We followed both reviewers' suggestions (see p. 6, l. 3-23).

- I agree on the use of declared data (instead of refunded damages) but it is not clear whether implemented damage data above 15.000 euros were verified or not. If this is the case, data below 15.000 euros are less reliable and authors should take this aspect into account in the analysis.

This part will be better clarified in the revised manuscript, in order to keep in consideration both Reviewers' comments.

Done (see p. 6, l. 8-11).

- I do not agree with the use of OMI data for the assessment of buildings value that, as stated by the authors in the Conclusions, "are more an expression of the overall economic well-being of a specific area" rather than of the real value of the buildings. (Re)construction costs are more suitable to the objective in my opinion.

We used the OMI values because they are one of the few reliable economic data that are available freely and homogeneously at a national level for provisional. Also, the use of these economic values is still deem to be informative for ex-ante damage estimation for planning activities. Moreover, reconstruction and restoration costs were not available when we started the analysis and the compilation of the dataset. Nevertheless, we will acknowledge this possibility in the revised manuscript.

We chose to keep the OMI values for the assessment of buildings values for the reasons explained above and we specified them in the revised manuscript (see p. 6, l. 28 – p. 7, l. 7).

Section 4.1

- The description of the damage models can be shortened by referring to available literature and leaving only the significant information for the paper (i.e. how models have been implemented).

Ok, thanks. We will shorten this description in the revised manuscript.

Done.

- Authors implement models developed to be applied at the micro-scale (e.g. MCM, Flemo-PS) and models developed to be applied at the meso-scale (e.g. Rhine Atlas, JRCs). I guess whether damage estimation (i.e. models' performance) is influenced by the different levels of knowledge/detail of input variables required by the models vs. available data. Did authors explore this aspect?

This aspect will be better discussed in the revised manuscript. We believe that this fact explains the differences among the performance of the models and the similar performances of the models at different scales. We will also take this opportunity to better strengthen the need for a more informed and rational selection of the damage model, which seldom appears to be the case in common practice, i.e. the level of detail of each input variable required by each model is always overlooked or neglected.

Done (see p. 15, l. 12-18 and p. 18, l. 26-31).

Section 4.1.1

- How authors converted the absolute curves of MCM in relative curves? MCM curves were developed in 2005 while the flood occurred in 2014; Did authors apply a discount rate to estimated damage? Why authors chose to convert absolute curves by mean of the average economic building value in the study area rather than by using different values for the different OMI zones? I would adopt this second option as MCM is a "micro" scale damage model.

Thanks, we will consider the possibility to apply the MCM curve as suggested.

We further investigated the economic trend of the Secchia study area building values between 2005 and 2014 and, mainly due to the recent economic crisis, the buildings' values did not vary substantially. For this reason, we neglected the application of a discount rate in the damages estimation (see p. 9, l. 17-18). In addition, according to previous studies and as better specified in the revised manuscript (see p. 9, l. 18-20), we considered a unique average economic value for the different OMI zones, being the values in all of them quite similar and for the sake of simplicity. The revised manuscript better illustrates the procedure to convert the absolute curve into relative values. (see p. 9, l. 14-17)

Section 4.2

- Which is the formulation of SEMP?

There is no formulation of the SEMP curve, because it comes out from the interpolation of the median damage values for each class (i.e. bin) of 25 cm water depth. We will better clarify this in the text that present the procedure to develop the model.

Done (see par. 4.2.1). In addition, we added a table with values in the Appendix (see Table A1).

Section 5.1

- From figures 7, 8, 9, it seems that uni-variable local models always estimate a relative damage around 0.1 (independently of the value of the dependent variable). Did authors notice that? How it can be justified?

We sincerely thank the Reviewer because his/her comment enabled us to identify a limitation of the previous study. Locally derived models consider an intercept different from zero, which we do not consider anymore to be realistic and representative of the buildings in the study area (i.e. additional direct verification enabled us to see that only a few affected buildings have a basement, whereas the norm is not to have any underground level for the impacted buildings). We are already working at the development of more robust empirical models, that have intercept equal to zero and we will present these models in the revised manuscript.

Done. We updated the results accordingly (see par. 4.2.2 and the results' chapter).

- How authors justify the bad performance of SVM in estimating the total absolute damage?

Thanks for this comment, which helped us realizing that the caption is rather misleading (and will be adjusted). We believe that the difference -and poorer performance- is associated with the fact that SVM is identified for relative damages and not for actual absolute damages in monetary terms. We will better investigate this aspect in the revised manuscript.

Done (see p. 15, l. 3-4) and the captions of Tables 4 and 5.

- With regard to existing models, I expect that models with the best performance underestimate the total damage (as citizens tend to overestimate damage during declaration). In fact, four of the six best models underestimate. Can authors comment on that?

The Reviewer raises a very interesting consideration which we will incorporate in the discussion section of the revised manuscript. Thanks.

Done (see p. 15, l. 5-9).

Section 5.2

- This section could be rewritten and improved to better explain the significance of results. Finding correspondence between authors' considerations and figures/tables is not straightforward at present.

- There is no correspondence between Figure 11 and its description in the text. Check also models acronym. Correspondence between test and figures is often lacking.

Thank you for these suggestions, this part will be improved following both observations.

Done.

Section 5.3

- The link between the performance in estimating damage to buildings and damage to contents is not so evident to me. Why SMV that is the one with the best performance in estimating damage to buildings is quite bad in estimating damage to contents?

We believe that the reason is that the regression curve for contents damages is derived starting from the structural damages to buildings and this relationship is not so strong itself. We will examine more in depth the explanation of these results, performing additional analyses if needed, and adding discussion of this aspect to the revised manuscript.

After modifications explained in the reply to the comment to the Section 5.1, the updated results show a better agreement with the results for damages to buildings. We believe that the reason of the small differences in the ranking of the models is that the regression curve for contents damages is derived starting from the structural damages to buildings and this relationship is not so strong itself. We discussed this aspect in the revised manuscript (see par. 5.3).

Conclusions

- The transferability of local models stated in the last part of the section should be better discussed previously in the paper. Two/three sentences highlighting this point can make conclusions more robust

Thank you for the advice. We will improve the revised manuscript accordingly.

Done (see p. 18, l. 15-18).

NB

Pay attention to be consistent in terminology. Authors use damage to "contents" and "content" interchangeably. I guess they are typos. The same can be state for model acronyms (e.g. SMV sometimes becomes MV).

We will pay attention to the typos in the revised manuscript.

Done, thanks.

Specific minor comments (which can increase the readability and clarity of the paper)

Section 1

Pg. 2 line 17 "flood risk is the combination of hazard (i.e. the probability of a flood event with a certain intensity to occur in a specific area and in a specific time period) and consequences, providing for instance information on the vulnerability, i.e. the type and number of elements affected by a given flood event, and how well they are able to resist" □ from this statement, I understand that consequences and vulnerability are the same "concept", please rephrase

Ok, thanks. We will improve this description.

We modified this part (see p. 2, l. 13-15).

[revised manuscript text omitted]

---

## Referee Report (RR1)

Journal: NHESS
Title: **Development and assessment of uni- and multi-variable flood loss models for Emilia-Romagna (Italy)**
Author(s): Carisi et al.
MS No.: NHESS-2017-342
MS Type: Research Article
**Iteration: Second review**

I really appreciate the efforts made by the authors in improving the paper that is now clearer and more robust. However, I think that some critical points need further clarification. I report these points below as major criticisms. I also add some minor comments that could improve the readability of the paper. I leave authors the choice to consider them or not. Finally, typos are highlighted.

P.S. I still do not agree on the use of the OMI values, as reconstruction costs can be easily derived from price lists. But, I accept this as a modelling choice.

**Major criticisms**

1. It is still not clear to me how the MCM model has been transformed into a relative model. Why authors refer to Secchia values and not UK values for the calculation of the relative curve? I would adopt market values referring to UK in 2005 not Italian ones. This is an important point to understand/explain estimations supplied by the model. Likewise, knowing whether relative damage in literature models has been derived by means of the reconstruction value or of the market value can support results discussion.

2. I do not understand the motivation given by authors on the worse performance of SMV with respect to other models, in terms of absolute damage estimation (i.e. that this is due to the fact that the model is identified for relative damage). Indeed, if the difference between relative and absolute damage is only due to the building value (which is considered by the model as non "important" variable) why such a behaviour should occur? I would like to have a more detailed explanation.

3. I also do not agree on the motivation supplied by authors on why empirically derived model underestimate damage, i.e. "looking at the empirically derived models, for example, the most precise model in terms of RMSE (SMV model) underestimates loss to buildings. This result can be expected and explained with the fact that citizens tend to overestimate damage during declaration and, consequently, observed loss is higher than estimated ones". This is true if models were derived from "correct" (i.e. not overestimated) data but the calibration dataset was made up by real "overestimated" data (as stated by authors). Please, discuss.

4. Pg. 15 line 12-18; "The reason behind this fact must be attributed to the morphologic and socio-economic context where this models have been drown, that differs considerably from the Secchia ones, in addition to the different criteria adopted to develop them. In fact, an other factor that influences the performances of the literature models applied on the Secchia case study is the different scale on which these curves are calibrated and applied: some of them are developed to be applied at the micro-scale (e.g. MCM, FLEMOps), while other at the meso-scale (e.g. Rhine Atlas, JRC curves). However, also among the meso-scale curves there are some of them with better results in estimating damages in the Secchia area than others, but it is difficult to identify a-priori which curve is better for a certain context." → I do not agree. I think that a deeper investigation on models properties and assumptions (e.g. hazard and vulnerability features of the context where they have been derived, values used for translating absolute damage into relative damage, level of aggregation of original data) can guide the identification of most suitable models. This should be discussed in the paper.

5. As regard transferability, I think that some considerations must be added on the role of vulnerability. Figure 10 shows that in the municipality of Bomporto and Bastiglia, despite different water depths, similar damage occurred. This can be explained by different vulnerability of buildings owning to the two dataset, which can also be the origin/ cause of prediction errors.

**Specific minor comments (which can increase the readability and clarity of the paper)**

Pg. 3 line 23-27; "The analyses described in this paper contribute to the understanding of possibilities and limitations of flood damage modelling in Northern Italy. In particular, we address the problem of lacking consistent data and the consequent difficulty in the development of reliable damage models for local applications. Also, our study investigates the open problem of transferability of empirical damage models to different areas and socio-economic contexts. Finally, the analysis aims to provide further insight on accuracy and robustness of uni- and multi-variable models in estimating flood loss to buildings and contents" → These aspects do not represent the real research questions of the paper, neither they are recalled and discussed in the final sections. I would suggest authors to remove the sentence.

Pg. 3 line 35; "As anticipated,….."→ where?

Pg. 4 line 26-29; Which is the southern boundary?

Pg. 5 line 4; "Thanks to several eyewitness accounts, video footage and studies conducted by the scientific committee" → which scientific committee? No reference is made to it before in the paper

Pg. 5 line 14-17; "Accordingly, citizens and property owners were asked to fill forms about public properties damages (Form A), private properties, furniture and registered goods damages (Form B), economic and productive activities damages (Form C) and agriculture and agro-industrial sector damages (Form D). In the present analysis, damage assessment focuses exclusively on private properties (Form B)" → name of the forms can be omitted as they have no sense for non-Italian readers.

Pg. 7 line 24- 26; The sentence contains a repetition, it can be simplified as "the reconstruction of the flood event was performed by means of Telemac-2D, a fully-2D hydrodynamic model which solves the 2D shallow water Saint Venant equations using the finite-element method within a computational mesh of triangular elements"

Pg. 8 line 32-33; "This section briefly recalls well known and largely employed literature depth-damage models, as well as two empirical depth-damage models and one multi-variable loss model that we identified "→ the models for Secchia are not recalled but derived in the research, please correct

Pg. 9 line 3-4; "All uni- and multi-variable models illustrated here are applied for predicting loss to household contents resulted from the January 2014 Secchia flood event"→ they are used also for buildings (i.e. structures and installations)

Pg. 11 line 4; "Usually, the buildings do not have an underground level. Therefore, for the impacted buildings a water depth equal to zero means no damages)." →This sentence repeats contents already discussed in the previous sentence and can be omitted.

Pg. 13 line 26-27; "…. is the possibility to understand the influence of the factors on the damage process for this specific context (different concept from the correlation one)"→ The difference in concepts is not so obvious for RF non-expert readers. Please, explain better.

Pg. 18 line 15-18; "Small differences in the ranking of the models, compared to Tables 4 and 5, is due to the fact that the regression curve for content damages is derived starting from the structural damages to

buildings and due to the variability of these values it brings this uncertainty also when applied for estimating content damages starting from the results of other models" → I do not understand what authors mean here. Please, rephrase; the sentence is not clear

Pg. 18 line 27-29; "Even though some literature models have similar performance to locally identified empirical models, the best performing literature models cannot be identified a-priori, which hampers the practical utilization of literature models themselves for predictive purposes" → see comment 4

Pg. 19 line 35-5; "According to Elmer et al. (2010), Schröter et al. (2014) and Schröter et al. (2016), the use of a number of explanatory variables to sustain more complex models (i.e., multi-variable model) leads to additional knowledge of the event, especially if the interdependence of the parameters are considered. However, this may introduce additional uncertainties, especially if the additional parameters are not collected specifically aiming at this kind of analysis. As a matter of fact, Secchia's database was collected for other purposes and does not include hydraulic parameters" → this sentence is not linked with previous or following one. A logical gap is present. I suggest to remove it.

Pg. 19 line 15-17 "According to Amadio et al. (2016), Molinari et al. (2012), Molinari et al. (2014b), and Scorzini and Frank (2015), the most urgent need in Italy, as far as loss estimation is concerned, is to identify guidelines, valid for the whole country, to collect consistent and comparable data, even if they relate to different context" → a proposal for a standardised procedure for data collection is included in Ballio et al., The RISPOSTA procedure for the collection, storage and analysis of high quality, consistent and reliable damage data in the aftermath of floods, *Journal of Flood Risk Management*, 2015

Pg. 19 line  24-25 "this study demonstrates that models can be transferred to similar contexts with satisfying results" → similar context on the bases of what? Please add

Pg. 19 line 6-7 "Finally, our study also emphasizes that loss-data collection is a fundamental and delicate task, and data-collection protocols are urgently needed for harmonizing and standardizing the compilation of flood-loss data sets" → the concept has been already state previously. The sentence can be removed.

**Figures and Tables**

Figure 3 - caption "Grey points in the background represent the observed relative loss (buildings only)" → the sentence is repeated

Figure 9 – Top panel is in Italian

Table A1 → it is not clear what values in the second column represent. Are they the mean bin value of observed damages? Please specify

**Typos**

Pg. 7 line 12; "this difference…." → I guess there is a typo. "this" must be replaced with "the"

Pg. 9 line 26. "The FLEMOps model assesses relative flood damages to private households referring us to several factors" → "us" must be deleted

Pg. 12 line 11 → "Depends of"  should be replaced with "Depends on"

**Bibliography**

I did not check the bibliography for lack of time

---

## Author Response (AR2)

**DEVELOPMENT AND ASSESSMENT OF UNI- AND MULTI-VARIABLE FLOOD LOSS MODELS FOR EMILIA-ROMAGNA (ITALY)**

by Francesca Carisi, Kai Schröter, Alessio Domeneghetti, Heidi Kreibich, Attilio Castellarin

MS No.: NHESS-2017-342; MS Type: Research Article; **Iteration: Second review**

**REPLY TO EDITORS' AND REVIEWERS' COMMENTS**

We would like to sincerely thank the Editor and the Reviewers for their useful comments and the challenge to improve the quality of the manuscript even more. Our re-revised manuscript addresses all observations and suggestions raised in the reports, by adding detailed explanations as required, in order to make our approach more robust and express our message even understandable.

Besides all the recommended modifications, we carefully revised the manuscript again, in order to correct errors and typos and to improve the readability and clarity of the paper.

The rest of the document uses the following notation:

- Black: original comments from Reviewers and Editor
- Red: actual revisions implemented in the re-revised manuscript, together with an explicit indication to the revised parts in the manuscript (i.e. pages and lines of the re-revised manuscript), when applicable.

**EDITOR DECISION:**

**Publish subject to minor revisions (review by editor)**
(06 Jun 2018) by Margreth Keiler

Comments to the Author:

Dear Francesca Carisi and co-authors,

Thank you very much for considering the comments of the referees. The referees and myself highly appreciate the efforts you invests to improve the manuscript. Both referees were interested to provide a second report on your manuscript.

Referee 1 only have two minor comments, however referee 2 recommends to add for some points a more detailed explanation, to discuss several aspects that may influence your results or contribute to uncertainties in the results (major criticisms). One of these aspects was also highlighted by referee 1.

I recommend to address these highlighted aspects in your manuscript and I am very positive that this will contribute to improve the understanding of your approach and the clear message of your manuscript.

I will review this minor revision of your manuscript.

I look forward to receiving the revised version of your manuscript soon.

Regards, Margreth Keiler
NHESS Editor
Associate Professor of Geomorphology, Natural Hazards and Risk Research, University of Bern

We would like to sincerely thank the Editor for her positive comment and input to further improve our manuscript.

**ANONYMOUS REFEREE #1**

The Authors answered all of my previous questions and considered the comments.
We would like to appreciatively thank the Anonymous Referee #1 for his/her positive judge and again for his/her suggestions in the previous review step, which helped us to focus on unclear and unprecise parts of the manuscript, improving the quality of the study.

Here are listed some last minor recommendations to improve the manuscript:

Figure 9: Please translate the Italian legend and axis description. Done.

Line 8-9, page 15: "This result can be expected and explained with the fact that citizens tend to overestimate damage during declaration and, consequently, observed loss is higher than estimated ones." This sentence may be misunderstood, please differentiate between estimations of the citizens and estimation of the model.
Thank you for your observation. We agree and revised this part accordingly, referring only to literature models' estimations compared with citizens' claims (see p. 15, l. 15-19).

**ANONYMOUS REFEREE #2**

I really appreciate the efforts made by the authors in improving the paper that is now clearer and more robust. However, I think that some critical points need further clarification. I report these points below as major criticisms. I also add some minor comments that could improve the readability of the paper. I leave authors the choice to consider them or not. Finally, typos are highlighted.
P.S. I still do not agree on the use of the OMI values, as reconstruction costs can be easily derived from price lists. But, I accept this as a modelling choice.
We would like to sincerely thank the Anonymous Referee #2 for his/her critical and careful analysis of our manuscript, that truly gave us the possibility to improve the clarity of the presentation.

**Major criticisms**

1. It is still not clear to me how the MCM model has been transformed into a relative model. Why authors refer to Secchia values and not UK values for the calculation of the relative curve? I would adopt market values referring to UK in 2005 not Italian ones. This is an important point to understand/explain estimations supplied by the model. Likewise, knowing whether relative damage in literature models has been derived by means of the reconstruction value or of the market value can support results discussion.
   We are very glad the Reviewer raised this comment, which enabled us to realize that the description we adopted in our revised manuscript (1st revision) was erroneous and absolutely

misleading. We actually adopted in our study the very MCM depth-damage curve reported in Jongman et al. (2012), which the authors relativized by referring to the original loss values in 2005 GBP. We clarify this point in our re-revised manuscript (see p. 9, l. 16-22).

2. I do not understand the motivation given by authors on the worse performance of SMV with respect to other models, in terms of absolute damage estimation (i.e. that this is due to the fact that the model is identified for relative damage). Indeed, if the difference between relative and absolute damage is only due to the building value (which is considered by the model as non "important" variable) why such a behaviour should occur? I would like to have a more detailed explanation.

We thank the Reviewer because this observation enabled us to improve the presentation of the results of our analysis. We adopted different metrics in order to analyze the performance of each model looking at different aspects. For example, RMSE looks at the difference between each estimation and the corresponding observed value in absolute terms (no sign), while the total estimated loss in monetary terms records whether the model under- or over-estimates the observed damages. Having most of the damage estimations small errors, their mean value expressed as RMSE is also small. Fig. 8 shows that, although the points lie close to the 1:1 line, most of them lie below it, i.e. SMV in general underestimates the observed damages. The sum of all the differences between observed and estimated loss results to be quite large, due to the fact that they are taken with their sign. We explained this aspect in the re-revised manuscript (see p. 14, l. 7-15; p. 14, l. 8 – p. 15, l. 7).

3. I also do not agree on the motivation supplied by authors on why empirically derived model underestimate damage, i.e. "looking at the empirically derived models, for example, the most precise model in terms of RMSE (SMV model) underestimates loss to buildings. This result can be expected and explained with the fact that citizens tend to overestimate damage during declaration and, consequently, observed loss is higher than estimated ones". This is true if models were derived from "correct" (i.e. not overestimated) data but the calibration dataset was made up by real "overestimated" data (as stated by authors). Please, discuss.

Thank you for raising another important comment on a part of the manuscript that needs to be further clarified. We modified this part in the re-revised manuscript (see p. 15, l. 15-19).

4. Pg. 15 line 12-18; "The reason behind this fact must be attributed to the morphologic and socio-economic context where this models have been drown, that differs considerably from the Secchia ones, in addition to the different criteria adopted to develop them. In fact, an other factor that influences the performances of the literature models applied on the Secchia case study is the different scale on which these curves are calibrated and applied: some of them are developed to be applied at the micro-scale (e.g. MCM, FLEMOps), while other at the meso-scale (e.g. Rhine Atlas, JRC curves). However, also among the meso-scale curves there are some of them with better results in estimating damages in the Secchia area than others, but it is difficult to identify a-priori which curve is better for a certain context." I do not agree. I think that a deeper investigation on models properties and assumptions (e.g. hazard and vulnerability features of the context where they have been derived, values used for translating absolute damage into relative

damage, level of aggregation of original data) can guide the identification of most suitable models. This should be discussed in the paper.

We thank the Reviewer for this suggestion. We included a deeper discussion in the re-revised manuscript (see p. 15, l. 23-29).

5. As regard transferability, I think that some considerations must be added on the role of vulnerability. Figure 10 shows that in the municipality of Bomporto and Bastiglia, despite different water depths, similar damage occurred. This can be explained by different vulnerability of buildings owning to the two dataset, which can also be the origin/ cause of prediction errors.

Thanks for this comment, we incorporated this observation in the re-revised manuscript (see p. 17, l. 8-10).

**Specific minor comments (which can increase the readability and clarity of the paper)**

Pg. 3 line 23-27; "The analyses described in this paper contribute to the understanding of possibilities and limitations of flood damage modelling in Northern Italy. In particular, we address the problem of lacking consistent data and the consequent difficulty in the development of reliable damage models for local applications. Also, our study investigates the open problem of transferability of empirical damage models to different areas and socio-economic contexts. Finally, the analysis aims to provide further insight on accuracy and robustness of uni- and multi-variable models in estimating flood loss to buildings and contents" These aspects do not represent the real research questions of the paper, neither they are recalled and discussed in the final sections. I would suggest authors to remove the sentence.

Actually, we prefer to keep this part, which we consider to be functional to the introduction of the study main objectives. Nevertheless, we improved its structure, better describing the real focus of the paper and taking into consideration the main conclusions (see p. 3, l. 33 – p. 4, l. 3).

Pg. 3 line 35; "As anticipated,….." where? Corrected.

Pg. 4 line 26-29; Which is the southern boundary? Specified.

Pg. 5 line 4; "Thanks to several eyewitness accounts, video footage and studies conducted by the scientific committee" which scientific committee? No reference is made to it before in the paper. Added.

Pg. 5 line 14-17; "Accordingly, citizens and property owners were asked to fill forms about public properties damages (Form A), private properties, furniture and registered goods damages (Form B), economic and productive activities damages (Form C) and agriculture and agro-industrial sector damages (Form D). In the present analysis, damage assessment focuses exclusively on private properties (Form B)" name of the forms can be omitted as they have no sense for non-Italian readers. Done.

Pg. 7 line 24- 26; The sentence contains a repetition, it can be simplified as "the reconstruction of the flood event was performed by means of Telemac-2D, a fully-2D hydrodynamic model which solves the 2D shallow water Saint Venant equations using the finite-element method within a computational mesh of triangular elements". Done.

Pg. 8 line 32-33; "This section briefly recalls well known and largely employed literature depth-damage models, as well as two empirical depth-damage models and one multi-variable loss model that we identified " the models for Secchia are not recalled but derived in the research, please correct. Done.

Pg. 9 line 3-4; "All uni- and multi-variable models illustrated here are applied for predicting loss to household contents resulted from the January 2014 Secchia flood event" they are used also for buildings (i.e. structures and installations). Corrected.

Pg. 11 line 4; "Usually, the buildings do not have an underground level. Therefore, for the impacted buildings a water depth equal to zero means no damages)." This sentence repeats contents already discussed in the previous sentence and can be omitted. Done.

Pg. 13 line 26-27; "…. is the possibility to understand the influence of the factors on the damage process for this specific context (different concept from the correlation one)" The difference in concepts is not so obvious for RF non-expert readers. Please, explain better. Done.

Pg. 18 line 15-18; "Small differences in the ranking of the models, compared to Tables 4 and 5, is due to the fact that the regression curve for content damages is derived starting from the structural damages to buildings and due to the variability of these values it brings this uncertainty also when applied for estimating content damages starting from the results of other models" I do not understand what authors mean here. Please, rephrase; the sentence is not clear. Done.

Pg. 18 line 27-29; "Even though some literature models have similar performance to locally identified empirical models, the best performing literature models cannot be identified a-priori, which hampers the practical utilization of literature models themselves for predictive purposes" see comment 4. Modified.

Pg. 19 line 35-5; "According to Elmer et al. (2010), Schröter et al. (2014) and Schröter et al. (2016), the use of a number of explanatory variables to sustain more complex models (i.e., multi-variable model) leads to additional knowledge of the event, especially if the interdependence of the parameters are considered. However, this may introduce additional uncertainties, especially if the additional parameters are not collected specifically aiming at this kind of analysis. As a matter of fact, Secchia's database was collected for other purposes and does not include hydraulic parameters" this sentence is not linked with previous or following one. A logical gap is present. I suggest to remove it. Done.

Pg. 19 line 15-17 "According to Amadio et al. (2016), Molinari et al. (2012), Molinari et al. (2014b), and Scorzini and Frank (2015), the most urgent need in Italy, as far as loss estimation is concerned, is to identify guidelines, valid for the whole country, to collect consistent and comparable data, even if they relate to different context" a proposal for a standardised procedure for data collection is included in Ballio et al., The RISPOSTA procedure for the collection, storage and analysis of high quality, consistent and reliable damage data in the aftermath of floods, *Journal of Flood Risk Management*, 2015. Added.

Pg. 19 line 24-25 "this study demonstrates that models can be transferred to similar contexts with satisfying results" similar context on the bases of what? Please add. Done.

Pg. 19 line 6-7 "Finally, our study also emphasizes that loss-data collection is a fundamental and delicate task, and data-collection protocols are urgently needed for harmonizing and standardizing the compilation of flood-loss data sets" the concept has been already state previously. The sentence can be removed. Done.

**Figures and Tables**

Figure 3 - caption "Grey points in the background represent the observed relative loss (buildings only)" the sentence is repeated. Deleted.

Figure 9 – Top panel is in Italian. Modified.

Table A1. it is not clear what values in the second column represent. Are they the mean bin value of observed damages? Please specify. Done.

**Typos**

Pg. 7 line 12. "this difference…." I guess there is a typo. "this" must be replaced with "the". Done.

Pg. 9 line 26. "The FLEMOps model assesses relative flood damages to private households referring us to several factors". "us" must be deleted. Done.

Pg. 12 line 11. "Depends of" should be replaced with "Depends on". Done.

**Bibliography**

I did not check the bibliography for lack of time

[revised manuscript text omitted]